# Maturing *Mycobacterium smegmatis* peptidoglycan requires non-canonical crosslinks to maintain shape

Catherine Baranowski[1], Michael A Welsh[2], Lok-To Sham[2,3], Haig A Eskandarian[4,5], Hoong Chuin Lim[2], Karen J Kieser[1], Jeffrey C Wagner[1], John D McKinney[4], Georg E Fantner[5], Thomas R Ioerger[6], Suzanne Walker[2], Thomas G Bernhardt[2], Eric J Rubin[1,2]*, E Hesper Rego[7]*

[1]Department of Immunology and Infectious Disease, Harvard TH Chan School of Public Health, Boston, United States; [2]Department of Microbiology and Immunobiology, Harvard Medical School, Boston, United States; [3]Department of Microbiology and Immunology, National University of Singapore, Singapore, Singapore; [4]School of Life Sciences, Swiss Federal Institute of Technology in Lausanne, Lausanne, Switzerland; [5]School of Engineering, Swiss Federal Institute of Technology in Lausanne, Lausanne, Switzerland; [6]Department of Computer Science and Engineering, Texas A&M University, Texas, United States; [7]Department of Microbial Pathogenesis, Yale University School of Medicine, New Haven, United States

*For correspondence:
erubin@hsph.harvard.edu (EJR);
hesper.rego@yale.edu (EHR)

Competing interests: The authors declare that no competing interests exist.

**Abstract** In most well-studied rod-shaped bacteria, peptidoglycan is primarily crosslinked by penicillin-binding proteins (PBPs). However, in mycobacteria, crosslinks formed by L,D-transpeptidases (LDTs) are highly abundant. To elucidate the role of these unusual crosslinks, we characterized *Mycobacterium smegmatis* cells lacking all LDTs. We find that crosslinks generate by LDTs are required for rod shape maintenance specifically at sites of aging cell wall, a byproduct of polar elongation. Asymmetric polar growth leads to a non-uniform distribution of these two types of crosslinks in a single cell. Consequently, in the absence of LDT-mediated crosslinks, PBP-catalyzed crosslinks become more important. Because of this, *Mycobacterium tuberculosis* (Mtb) is more rapidly killed using a combination of drugs capable of PBP- and LDT- inhibition. Thus, knowledge about the spatial and genetic relationship between drug targets can be exploited to more effectively treat this pathogen.
DOI: https://doi.org/10.7554/eLife.37516.001

## Introduction

Peptidoglycan (PG) is an essential component of all bacterial cells (*Vollmer et al., 2008a*), and the target of many antibiotics. PG consists of linear glycan strands crosslinked by short peptides to form a continuous molecular cage surrounding the plasma membrane. This structure maintains cell shape and protects the plasma membrane from rupture. Our understanding of PG is largely derived from studies on laterally growing model rod-shaped bacteria like *Escherichia coli* and *Bacillus subtilis* (*Figure 1—figure supplement 1A*). In these organisms, new PG is constructed along the lateral side wall by the concerted effort of glycosyltransferases, which connect the glycan of a new PG subunit to the existing mesh, and transpeptidases, which link peptide side chains. An actin-like protein, MreB, positions this multi-protein complex along the short axis of the cell so that glycan strands are inserted circumferentially, creating discontinuous hoops of PG around the cell (*Domínguez-*

**eLife digest** Most bacteria have a cell wall that protects them and maintains their shape. Many of these organisms make their cell walls from fibers of proteins and sugars, called peptidoglycan. As bacteria grow, peptidoglycan is constantly broken down and reassembled, and in many species, new units of peptidoglycan are added into the sidewall. However, in a group of bacteria called mycobacteria, which cause tuberculosis and other diseases, the units are added at the tips.

The peptidoglycan layer is often a successful target for antibiotic treatments. But, drugs that treat tuberculosis do not attack this layer, partly because we know very little about the cell walls of mycobacteria.

Here, Baranowski et al. used genetic manipulation and microscopy to study how mycobacteria build their cell wall. The results showed that these bacteria link peptidoglycan units together in an unusual way. In most bacteria, peptidoglycan units are connected by chemical links known as 4-3 crosslinks. This is initially the same in mycobacteria, but as the cell grows and the cell wall expands, these bonds break and so-called 3-3 crosslinks form. In genetically modified bacteria that could not form these 3-3 bonds, the cell wall became brittle and weak, and the bacteria eventually died.

These findings could be important for developing new drugs that treat infections caused by mycobacteria. Baranowski et al. demonstrate that a combination of drugs blocking both 4-3 and 3-3 crosslinks is particularly effective at killing the bacterium that causes tuberculosis.

DOI: https://doi.org/10.7554/eLife.37516.002

*Escobar et al., 2011*; *Garner et al., 2011*). This orientation of PG creates a mechanical anisotropy that is responsible for rod shape (*Hussain et al., 2018*).

However, not all rod-shaped bacteria encode MreB. In fact, there are important differences between model bacteria and Actinobacteria like mycobacteria, a genus of rod-shaped bacteria that includes the major human pathogen *Mycobacterium tuberculosis* (Mtb). In mycobacteria, new PG is inserted at the cell poles (at unequal amounts based on pole age), rather than along the lateral walls (*Figure 1A*). Additionally, mycobacteria are missing several factors, including MreB, that are important for cell elongation (*Kieser and Rubin, 2014*). Furthermore, in *E. coli* and *B. subtilis* the vast majority (>90%) of the peptide linkages are created by D,D-transpeptidases known as penicillin-binding proteins (PBPs) (*Pisabarro et al., 1985*). PBPs, the targets of most β-lactams, link the fourth amino acid of one peptide side chain to the third amino acid of another, forming 4–3 crosslinks. Peptidoglycan crosslinks can also be catalyzed by L,D-transpeptidases (LDTs), which link peptide side chains by the third amino acid forming 3–3 linkages (*Figure 1—figure supplement 1B*). In mycobacteria, these 3–3 crosslinks, are highly abundant, accounting for at least 60% of linkages (*Kumar et al., 2012*; *Lavollay et al., 2008*; *Wietzerbin et al., 1974*). Although there has been extensive characterization of LDTs *in vitro* (*Cordillot et al., 2013*; *Dubée et al., 2012*; *Lavollay et al., 2008*; *Magnet et al., 2007*; *Mainardi et al., 2005*; *Mainardi et al., 2007*; *Triboulet et al., 2013*), because PG has been most well studied in bacteria where 3–3 crosslinks are rare, the cellular role of these enzymes and the linkages they create is poorly understood. As is the case with PBPs, there exists many copies of LDTs in the cell - there are five LDTs in Mtb and six in *Mycobacterium smegmatis* (Msm), a non-pathogenic relative of Mtb (*Sanders et al., 2014*), making genetic characterization challenging. Also similarly to PBPs, LDT homologues do not appear to functionally overlap completely (*Cordillot et al., 2013*; *Kumar et al., 2017*; *Schoonmaker et al., 2014*).

Tuberculosis remains an enormous global health problem, in part, because treating even drug susceptible disease is difficult. The standard regimen includes a cocktail of four drugs given over six months. Treatment of drug-resistant Mtb is substantially longer and includes combinations of up to seven drugs (*Global Tuberculosis Report, 2017*). While some of the most important anti-mycobacterials target cell wall synthesis, surprisingly, drugs that target PG are not part of the core treatment for either drug-susceptible or drug-resistant disease. However, carbapenems, β-lactam antibiotics that potently inhibit LDTs *in vitro* (*Cordillot et al., 2013*; *Dubée et al., 2012*; *Lavollay et al., 2008*; *Mainardi et al., 2007*; *Triboulet et al., 2013*), are also effective against drug resistant Mtb *in vitro* and drug-sensitive Mtb in patients (*Diacon et al., 2016*; *Hugonnet et al., 2009*).

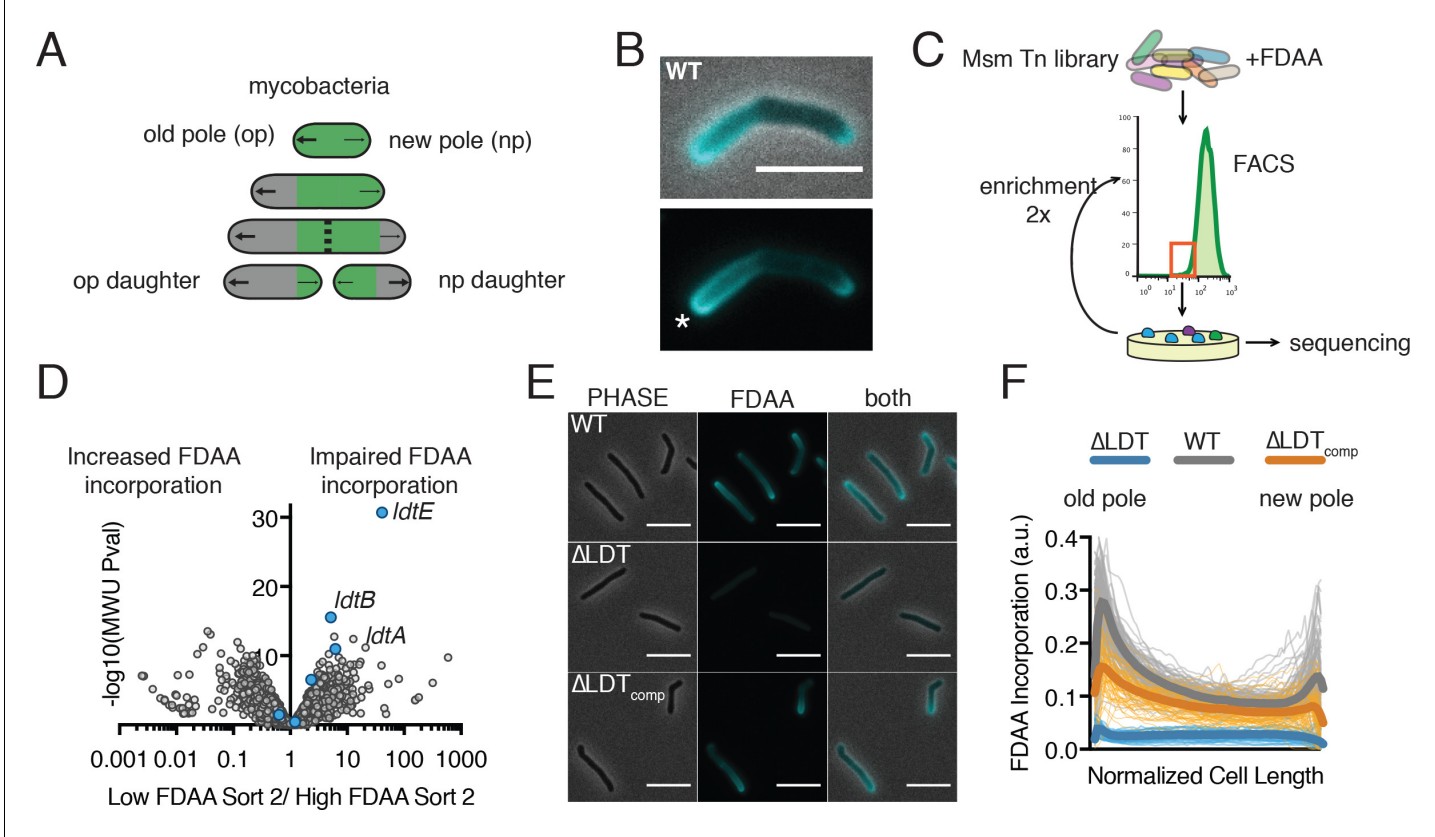

**Figure 1.** FDAAs are incorporated asymmetrically by L,D-transpeptidases. (**A**) Schematic of mycobacterial asymmetric polar growth. Green, old cell wall; grey, new material; dotted line, septum; large arrows, old pole growth; small arrows, new pole growth. (**B**) FDAA incorporation in log-phase WT Msm cell after 2 min incubation. Scale bar = 5 μm. Old pole marked with (*). (**C**) Schematic of Fluorescence Activated Cell Sorting (FACS)-based FDAA transposon library enrichment. An Msm transposon library was stained with FDAAs, the dimmest and brightest cells were sorted, grown, sorted again to enrich for transposon mutants that are unable or enhanced for FDAA incorporation. (**D**) Results from 1C screen. For each gene, the contribution to low or high staining population was calculated from transposon reads per gene. Plotted is the ratio of the population contribution from the second sort of low FDAA staining (L2) over the second sort of high FDAA staining (H2) cells compared to the Mann-Whitney *U* p-value. (**E**) Representative image of FDAA incorporation in log-phase WT, ΔLDT and ΔLDT$_{comp}$ cells. Scale bar = 5 μm. (**F**) Profiles of FDAA incorporation in log-phase WT (N = 98), ΔLDT (N = 40), and ΔLDT$_{comp}$ (N = 77) cells. Thick lines represent mean incorporation profile, thin lines are FDAA incorporation in single cells.

DOI: https://doi.org/10.7554/eLife.37516.003

The following source data and figure supplements are available for figure 1:

**Source data 1.** FDAA FACs screen data used for *Figure 1D*.
DOI: https://doi.org/10.7554/eLife.37516.009

**Source data 2.** FDAA incorporation distribution data used for *Figure 1F*.
DOI: https://doi.org/10.7554/eLife.37516.010

**Figure supplement 1.** Peptidoglycan synthesis overview.
DOI: https://doi.org/10.7554/eLife.37516.004

**Figure supplement 2.** Time-lapse microscopy maps FDAA incorporation pattern to old and new poles.
DOI: https://doi.org/10.7554/eLife.37516.005

**Figure supplement 2—source data 1.** Fluorescence values for panels C and E.
DOI: https://doi.org/10.7554/eLife.37516.006

**Figure supplement 3.** Fluorescent D-amino acid screen validation.
DOI: https://doi.org/10.7554/eLife.37516.007

**Figure supplement 4.** 3–3 crosslinks are not detectable in ΔLDT cells.
DOI: https://doi.org/10.7554/eLife.37516.008

But, why are LDTs important in mycobacteria? To explore this, we constructed a strain of Msm that lacks the ability to form 3–3 crosslinks. We find that 3–3 crosslinks are formed in maturing peptidoglycan and that they are necessary to stabilize the cell wall and prevent lysis. Cells that lose the ability to synthesize 3–3 crosslinks have increased dependence on 4–3 crosslinking. Thus, simultaneous inhibition of both processes results in rapid cell death.

## Results

### Fluorescent D-amino acids are incorporated asymmetrically by L,D-transpeptidases

PG uniquely contains D-amino acids, which can be conjugated to fluorescent probes (fluorescent D-amino acids, FDAAs) to visualize PG synthesis in live bacterial cells (*Kuru et al., 2012*). When we incubated Msmwith FDAAs for a short 2 min pulse ( < 2% of Msm's generation time) we observed incorporation at both poles, the sites of new PG insertion in mycobacteria (*Figure 1A,B*) (*Aldridge et al., 2012*). However, we also saw a gradient of fluorescence along the sidewalls, extending from the old pole (the previously established growth pole) that fades to a minimum at roughly mid-cell as it reaches the new pole (the pole formed at the last cell division) (*Figure 1B*, *Figure 1—figure supplement 2*).

To identify the enzymes responsible for this unexpected pattern of lateral cell wall FDAA incorporation, we performed a fluorescence-activated cell sorting (FACS)-based transposon screen (*Figure 1C*). Briefly, we stained an Msm transposon library with FDAA and repeatedly sorted the least fluorescent 12.5% of the population by FACS. After each sort we regrew cells, extracted gDNA and used deep sequencing to map the location of the transposons found in the low-staining population.

From this screen, we identified three LDTs (*ldtA - MSMEG_3528, ldtB - MSMEG_4745, ldtE - MSMEG_0233*) (*Figure 1D*) that appeared primarily responsible for FDAA incorporation. Deleting these three LDTs significantly reduced FDAA incorporation and this defect in incorporation could be partially complemented with constitutive expression of LdtE alone (*ldtE*-mRFP, *Figure 1—figure supplement 3A*). To further investigate the physiological role of LDTs, we constructed a strain lacking all 6 LDTs ($\Delta ldtAEBCGF$, hereafter $\Delta$LDT). Whole genome sequencing verified all six deletions and did not detect crossover events or chromosomal duplications (see supplemental methods). FDAA incorporation and 3–3 crosslinking are both nearly abolished in $\Delta$LDT cells and can be partially restored by complementation with a single LDT (*ldtE*-mRFP; $\Delta$LDT$_{comp}$) (*Figure 1E,F*, *Figure 1—figure supplements 3B* and *4*). Thus, as might be the case in *Bdellovibrio* (*Kuru et al., 2017*), FDAA incorporation in Msm is primarily LDT-dependent. LDTs have previously been shown to exchange non-canonical D-amino acids onto PG tetrapeptides in *Vibrio cholera* (*Cava et al., 2011*).

### 3–3 crosslinks are required for rod shape maintenance at aging cell wall

As deletion of a subset of LDTs in Msm produces morphologic changes (*Sanders et al., 2014*), we visualized $\Delta$LDT cells by time-lapse microscopy. We observed that a subpopulation of cells loses rod shape progressively over time, resulting in localized spherical blebs (*Figure 2A* – top row, *Figure 2—figure supplement 1A*, *Figure 2—video 1*). Complemented cells are able to maintain rod shape (*Figure 2—figure supplement 1B*). We reasoned that localized loss of rod shape may occur for two reasons: (1) spatially-specific loss of cell wall integrity and/or (2) cell wall deformation due to uncontrolled, local PG synthesis. If the first hypothesis were true, high osmolarity should protect cells against forming blebs. Indeed, switching cells from iso- to high- osmolarity prevented bleb formation over time (*Figure 2A* – bottom row, *Figure 2—video 2*). To test the second hypothesis, we stained $\Delta$LDT or WT cells with an amine-reactive dye, and observed outgrowth of new, unstained material (*Figure 2B*). Blebs that formed in the $\Delta$LDT cells retained stain, indicating a lack of new cell wall synthesis in the region. WT cells maintained rod shape over time at the stained portion of the bacillus. Collectively, these results indicate that 3–3 crosslinks are required to counteract turgor pressure and maintain rod shape in Msm. This led us to hypothesize that bleb formation is a result of a local defect in cell wall rigidity.

To directly measure cell wall rigidity, we used atomic force microscopy (AFM) on live $\Delta$LDT and WT cells. We measured the rigidity of cells in relation to their height. Generally, WT cells are stiffer

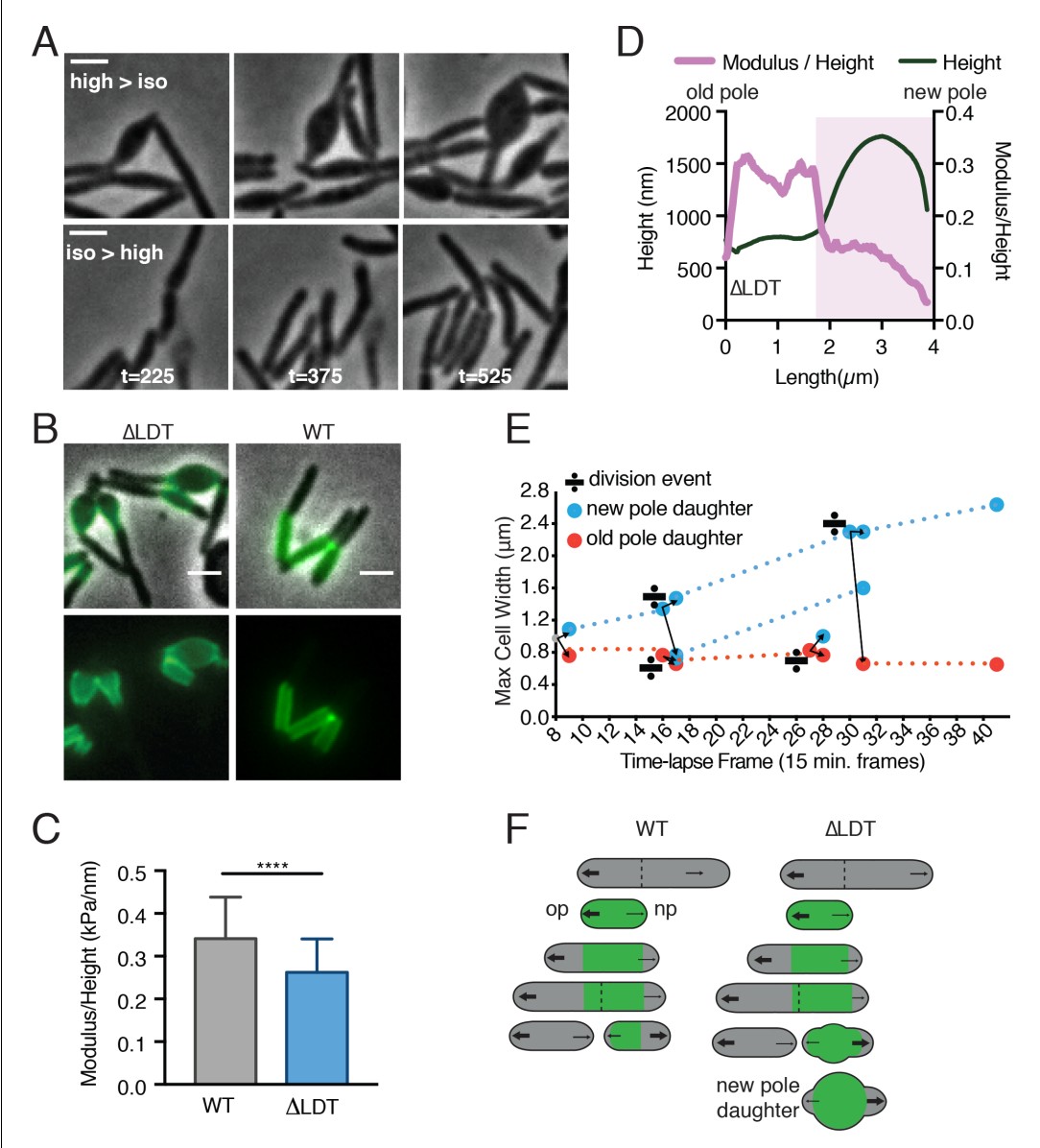

**Figure 2.** 3–3 crosslinks are required for rod shape maintenance at aging cell wall. (A) Msm ΔLDT time-lapse microscopy of cells switched from high- to iso- osmolar media (top row, see *Figure 2—video 1*), or iso- to high osmolar media (bottom row, see *Figure 2—video 2*). (high = 7H9+150 mM sorbitol; iso = 7H9). t = time in minutes post-osmolarity switch. (B) ΔLDT or WT cells were stained with Alexa 488 NHS-ester (green) to mark existing cell wall, washed, and visualized after outgrowth (unstained material). A, B scale bar = 2 μm. (C) Mean stiffness of WT (N = 73) and ΔLDT (N = 47) Msm cells as measured by atomic force microscopy. Mann-Whitney U p-Value ****<0.0001. (D) Representative profile of cell height and height-normalized stiffness (modulus/height) in a single ΔLDT cell. Pink-shaded portion highlights location of a bleb. (E) Maximum cell width of ΔLDT cell lineages over time. Width of new pole daughters = blue circle; width of old pole daughters = orange circle. Division signs denote a division event. At each division, there are two arrows from the dividing cell leading to the resulting new and old pole daughter cell widths (blue and orange respectively). (F) Model of rod shape loss in old cell wall of ΔLDT cells compared to WT. Green portions of the cell represents old cell wall; grey portion represents new cell wall. The larger arrows indicate more growth from the old pole, while smaller arrows show less relative growth from the new pole. Dotted lines represent septa. op = old pole, np = new pole.

DOI: https://doi.org/10.7554/eLife.37516.011

The following video, source data, and figure supplements are available for figure 2:

**Source data 1.** Modulus (kPa)/Height (nm) for WT and ΔLDT cells used for *Figure 2C*.
DOI: https://doi.org/10.7554/eLife.37516.014

**Source data 2.** Modulus and height for the representative ΔLDT cell corresponding to *Figure 2D*.
DOI: https://doi.org/10.7554/eLife.37516.015

*Figure 2 continued on next page*

*Figure 2 continued*

**Figure supplement 1.** ΔLDT cell morphological characteristics.
DOI: https://doi.org/10.7554/eLife.37516.012
**Figure supplement 2.** Inheritance of old cell wall and occurrence of blebs in new pole daughter cells.
DOI: https://doi.org/10.7554/eLife.37516.013
**Figure 2—video 1.** Time-lapse of ΔLDT cells in iso-osmolar media.
DOI: https://doi.org/10.7554/eLife.37516.016
**Figure 2—video 2.** Time-lapse of ΔLDT cells in high-osmolar media.
DOI: https://doi.org/10.7554/eLife.37516.017

than ΔLDT cells (*Figure 2C*). Blebs in ΔLDT cells can be identified by a sharp increase in height (*Figure 2D*, *pink shaded*). Since circumferential stress of the rod measured by AFM is proportional to the radius of the cell, and inversely proportional to the thickness of the cell wall (an immeasurable quantity by AFM), we used cell height, a proxy for radius, to normalize the stiffness measurement. We found that stiffness drops in the area of blebs (*Figure 2D*, *pink shaded*).

Why does loss of rod shape occur locally and only in a subpopulation of cells? Mycobacterial polar growth and division results in daughter cells with phenotypic differences (*Aldridge et al., 2012*). For example, the oldest cell wall is specifically inherited by the new pole daughter (*Figure 2—figure supplement 2A*, *Aldridge et al., 2012*). We hypothesized that the loss of rod shape might occur in specific progeny generated by cell division. Indeed, the daughter which inherited the new pole from the previous round of division, and the oldest cell wall, consistently lost rod shape over time, while the old pole daughter maintained rod shape (*Figure 2E*, *Figure 2—figure supplement 2B*). In addition, blebs localized to the oldest cell wall (*Figure 2B*), as visualized by pulse-chase labeling of the cell wall. Thus, 3–3 crosslinking is likely occurring in the oldest cell wall, which is non-uniformly distributed along a single cell and in the population via asymmetric polar growth and division. Taken together, these data suggest that LDTs act locally to reinforce aging PG and to maintain rod shape in a subpopulation of Msm cells - specifically, new pole daughters (*Figure 2F*).

## *Mycobacterium smegmatis* is hypersensitive to PBP inactivation in the absence of LDTs

Our observations lead to the following model: 4–3 crosslinks made by PBPs are formed at the poles where new PG is inserted and where pentapeptide substrates reside. These newly synthesized 4–3 crosslinks can then be gradually cleaved (by D,D-endopeptidases) as PG ages and moves toward the middle of the cell, leaving tetrapeptide substrates for LDTs to create 3–3 crosslinks. This is consistent with the FDAA incorporation pattern, which reflects the abundance of tetrapeptide substrates available for LDT exchange. Specifically, there are more available tetrapeptides near the poles and fewer near mid-cell, the site of older PG (*Figure 1F*). In the absence of LDTs to catalyze 3–3 crosslinks, old cell wall loses integrity and turgor pressure causes bleb formation.

This model predicts that ΔLDT cells should be even more dependent on 4–3 crosslinking than wild-type cells. To test this hypothesis, we used TnSeq (*Long et al., 2015*) to identify genes required for growth in cells lacking LDTs (*Figure 3A*). We found that mutants of two PBPs, *pbpA* (*MSMEG_0031* c) and *ponA2* (*MSMEG_6201*), were recovered at significantly lower frequencies in ΔLDT cells (*Figure 3B*). Likewise, using allele swapping (*Kieser et al., 2015b*) (*Figure 3C*, *Figure 3—figure supplement 1*), a technique that tests the ability of various alleles to support viability, we found that the transpeptidase (TP) activity of PonA1, which is non-essential in WT cells (*Kieser et al., 2015b*), becomes essential in ΔLDT cells (*Figure 3D*). Thus, cells that lack 3–3 crosslinks are more dependent on 4–3 crosslinking enzymes.

## Peptidoglycan synthesizing enzymes localize to differentially aged cell wall

Given our model, we hypothesized that enzymes catalyzing and processing different types of crosslinks should be differentially localized along the length of the cell. Specifically, we postulated that 4–3 generating PBPs would localize at sites of new PG, while 4–3 cleaving D,D-endopeptidases and 3–3 crosslinking LDTs would localize to sites of older PG. Polar growth segregates newer PG to the poles, and, as growth occurs, older PG migrates towards the middle of the cell. To test whether 4–3

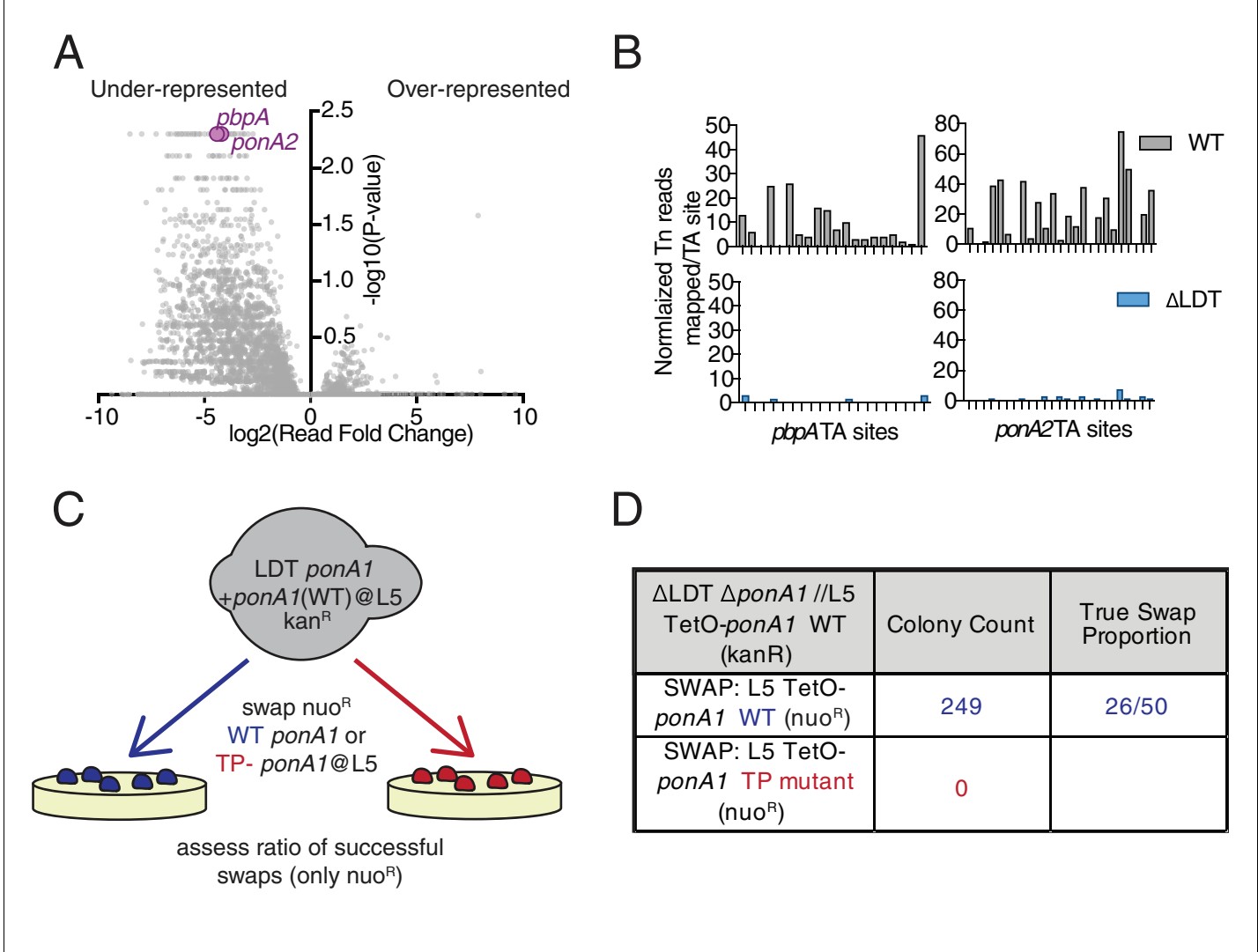

**Figure 3.** *Mycobacterium smegmatis* is hypersensitive to PBP inactivation in the absence of LDTs. (**A**) Fold change in the number of reads for transposon insertion counts in ΔLDT cells compared to WT Msm. p-value is derived from a rank sum test (*DeJesus et al., 2015*). (**B**) Transposon insertions per TA dinucleotide in *pbpA* and *ponA2* in WT (grey) and ΔLDT (blue) cells. (**C**) Schematic of L5 allele swapping experiment. (**D**) Results of WT or transpeptidase null *ponA1* allele swapping experiment in ΔLDT cells.

DOI: https://doi.org/10.7554/eLife.37516.018

The following source data and figure supplement are available for figure 3:

**Source data 1.** ΔLDT Tnseq data used for *Figure 3A*.
DOI: https://doi.org/10.7554/eLife.37516.020
**Source data 2.** Read counts per TA site in WT and ΔLDT cells for *ponA2* and *pbp2* used in *Figure 3B*.
DOI: https://doi.org/10.7554/eLife.37516.021
**Figure supplement 1.** L5 allele swapping to test essentiality of PonA1's ability to form 4 to 3 crosslinks (transpeptidation).
DOI: https://doi.org/10.7554/eLife.37516.019

and 3–3 crosslinking enzymes localize differently, we visualized fluorescent fusions of a PBP (PonA1), and an LDT (LdtE), (*Figure 4A*). Intriguingly, both enzymes localized in a gradient pattern along the long axis of the cell, not unlike the pattern observed for FDAA incorporation. We found that the distribution pattern of PonA1-RFP was highest at the old and new poles, where new PG is inserted (*Figure 4A,B, Figure 4—video 1, Figure 4—figure supplement 1A*). Compared to PonA1-RFP, the LdtE-mRFP localization is highest farther from the poles, more inward from the ends of the bacillus (albeit in a similar gradient pattern), at the sites of older PG (*Figure 4A,B, Figure 4—video 2,*

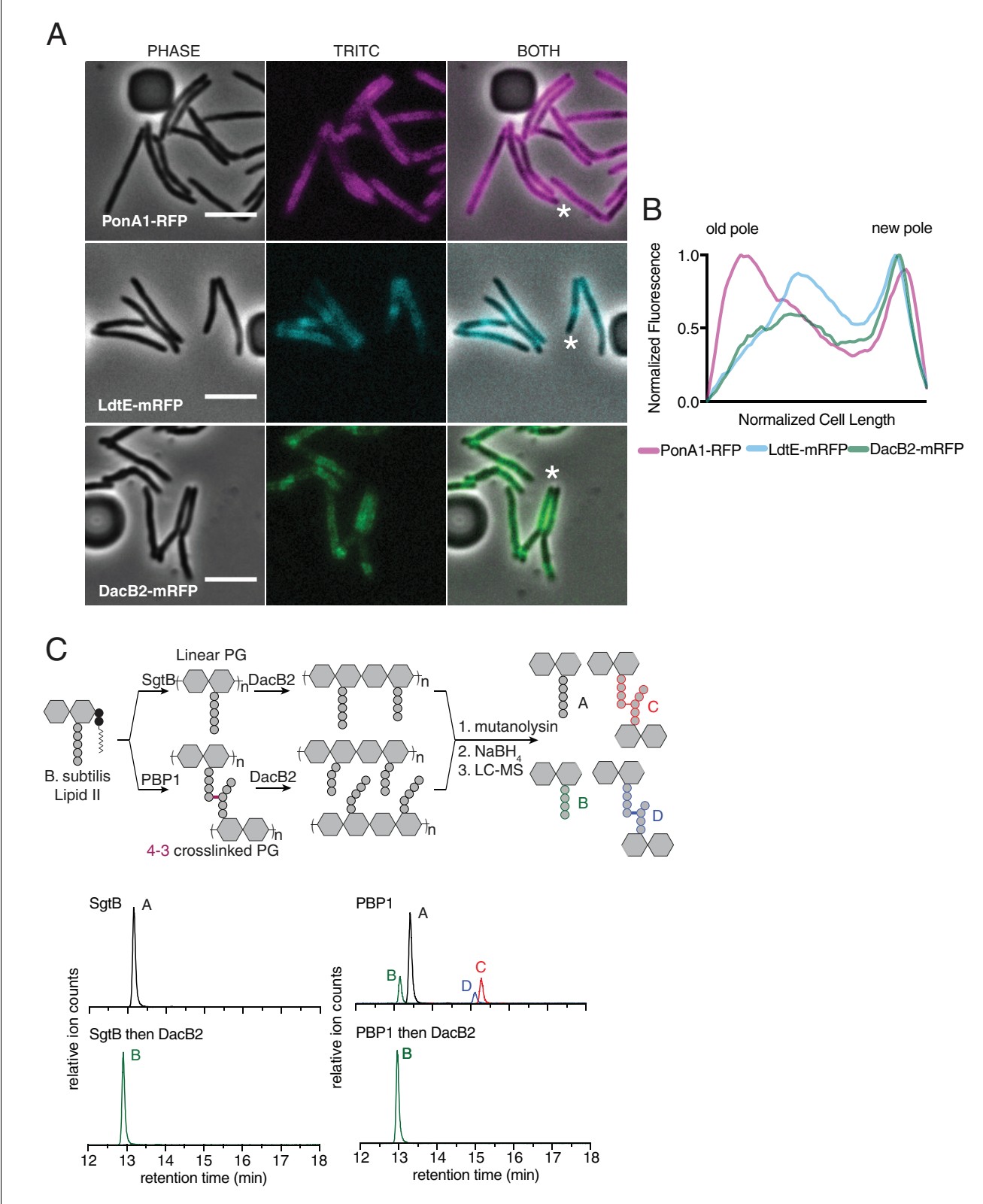

**Figure 4.** Peptidoglycan synthesizing enzymes localize to differentially aged cell wall. (**A**) Representative fluorescence image of PonA1-RFP (magenta, see *Figure 4—video 1*), LdtE-mRFP (cyan, see *Figure 4—video 2*), and DacB2-mRFP (green, see *Figure 4—video 3*). Scale bars = 5 μm. (**B**) Average PonA1-RFP (N = 24), LdtE-mRFP (N = 23) or DacB2-mRFP (N = 23) distribution in cells before division. (**C**) Schematic of the *in vitro* experiment to test D, D-carboxy- and D,D-endopeptidase activity of DacB2 (top). Lipid II extracted from *B. subtilis* is first polymerized into linear (using SgtB) or crosslinked

*Figure 4 continued on next page*

*Figure 4 continued*

(using *B. subtilis* PBP1) peptidoglycan and then reacted with DacB2. The reaction products are analyzed by LC-MS. Extracted ion chromatograms of the reaction products produced by incubation of DacB2 with peptidoglycan substrates (bottom).

DOI: https://doi.org/10.7554/eLife.37516.022

The following video, source data, and figure supplements are available for figure 4:

**Source data 1.** Fluorescence distributions used for *Figure 4B*.

DOI: https://doi.org/10.7554/eLife.37516.028

**Figure supplement 1.** PG synthetic enzyme localization at birth, 30, 60 and 90 min post-birth and at the frame before division (pre-division) in a representative cell.

DOI: https://doi.org/10.7554/eLife.37516.023

**Figure supplement 1—source data 1.** Fluorescence distributions used for *Figure 4—figure supplement 1A–C*.

DOI:

**Figure supplement 2.** MSMEG_2433 (DacB2) functions as a D,D-carboxypeptidase and D,D-endopeptidase in vitro.

DOI: https://doi.org/10.7554/eLife.37516.025

**Figure supplement 3.** CRISPRi knock-down of *dacB2* in Msm lacking LDTs reduces bleb size.

DOI: https://doi.org/10.7554/eLife.37516.026

**Figure supplement 3—source data 1.** Measurements of bleb width for *Figure 4—figure supplement 3*.

DOI: https://doi.org/10.7554/eLife.37516.027

**Figure 4—video 1.** Time-lapse of ponA1-RFP.

DOI: https://doi.org/10.7554/eLife.37516.029

**Figure 4—video 2.** Time-lapse of ldtE-mRFP.

DOI: https://doi.org/10.7554/eLife.37516.030

**Figure 4—video 3.** Time-lapse of dacB2-mRFP.

DOI: https://doi.org/10.7554/eLife.37516.031

*Figure 4—figure supplement 1B*). Thus, enzymes responsible for 4–3 and 3–3 crosslinks show distinctive subcellular localizations with respect to the site of new PG synthesis. This is consistent with the model that these enzymes act on differentially aged PG.

We next sought to localize a D,D-endopeptidase. As no D,D-endopeptidase has been clearly identified in mycobacteria, we used HHPRED (*Zimmermann et al., 2018*) to find candidates. By homology to the *E. coli* protein AmpH, an enzyme with both D,D- carboxy- and endopeptidase activity (*González-Leiza et al., 2011*), we identified DacB2 (MSMEG_2433), a protein previously shown to have D,D-carboxypeptidase activity in Msm (*Bansal et al., 2015*), as a candidate to also harbor D,D-endopeptidase capability. We expressed and purified DacB2 and found that it, like AmpH, had both D,D-carboxypeptidase and D,D-endopeptidase activity on peptidoglycan substrates generated *in vitro* (*Figure 4C*, *Figure 4—figure supplement 1A–C*). We used a recently developed CRISPRi system for mycobacteria to knockdown *dacB2* expression in ΔLDT cells (*Rock et al., 2017*). Induction of the sgRNA and dCas9 by anhydro-tetracycline (aTc) led to smaller blebs (*Figure 4—figure supplement 3*). Furthermore, DacB2-mRFP localized closer to LDT-mRFP, farther from the poles, at sites of older PG (*Figure 4A,B*, *Figure 4—video 3*, *Figure 4—figure supplement 1C*). Taken together, these data are consistent with a model in which blebs are formed in ΔLDT cells due to unchecked D,D-endopeptidase activity. Given that bleb formation is not completely rescued by knockdown of *dacB2*, we speculate that there are additional D,D-endopeptidases in *M. smegmatis*.

## Drugs targeting both PBPs and LDTs synergize to kill *Mycobacterium tuberculosis*

The importance of 3–3 crosslinks in mycobacteria suggests a unique vulnerability. While Mtb can be killed by most non-carbapenem (N-C) β-lactams like amoxicillin, which largely target the PBPs, carbapenem β-lactams, which target both PBPs and LDTs (*Kumar et al., 2017*; *Mainardi et al., 2007*; *Papp-Wallace et al., 2011*) are also effective against Mtb (*Diacon et al., 2016*; *Hugonnet et al., 2009*). It has been previously proposed (*Gonzalo and Drobniewski, 2013*; *Gupta et al., 2010*; *Kumar et al., 2017*; *Mainardi et al., 2007*) that more rapid killing of Mtb could be achieved with drug combinations that target both PBPs and LDTs. Msm Tnseq data suggests that typically dispensable 4–3 transpeptidase activity becomes essential in cells lacking LDTs (*Figure 3*), supporting the

notion that inhibition of both PBPs and LDTs could kill mycobacteria very successfully. Interestingly, while we could create a strain of Msm lacking all LDTs, previously published Mtb Tnseq data suggests that LDTs may be essential in the pathogen (*Kieser et al., 2015a*).

We utilized Msm and Mtb strains expressing the *luxABCDE* operon from *Photorhabdus luminescens* (*Andreu et al., 2012*; *Andreu et al., 2010*), where light production can be correlated to growth (*Figure 5—figure supplement 1*), to test if the combination of amoxicillin (a penam) and meropenem (a carbapenem) killed Msm or Mtb more rapidly than either drug alone. We found that these drugs together kill both Msm and Mtb faster than either alone (*Figure 5A,B*). Furthermore, this combination exhibits synergism in minimal inhibitory concentration in Mtb but, not against Msm (where synergism is defined as Σ Fractional Inhibitory Concentration <0.5 ('Synergism Testing: Broth Microdilution Checkerboard and Broth Macrodilution Materials and methods,' 2016), *Figure 5B*, *Figure 5—figure supplement 2*). This may reflect a difference in LDT expression or essentiality between Msm and Mtb.

## Discussion

The success of antibiotics that target PG, like β-lactams, has led to decades of research on this critical bacterial polymer. Recently developed fluorescent probes (FDAAs) have been used extensively to study PG synthesis in live cells of numerous bacterial species (*Kuru et al., 2012*; *Kuru et al., 2017*; *Liechti et al., 2014*). Intriguingly, these probes can be incorporated through diverse pathways in different bacteria and thus, their pattern can mark distinct processes (*Kuru et al., 2012*). We find that in mycobacteria, FDAA incorporation is primarily LDT-dependent. FDAA incorporation in Msm shows an unusual gradient pattern (*Botella et al., 2017*), suggesting an asymmetric distribution of

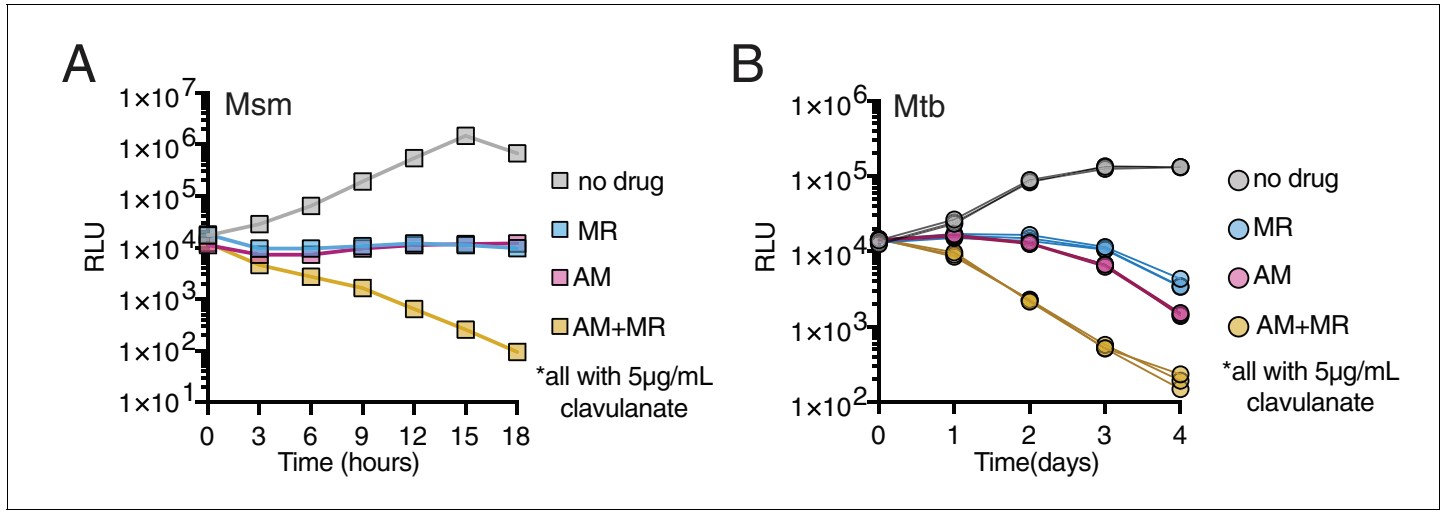

**Figure 5.** Drugs targeting both PBPs and LDTs kill mycobacteria more rapidly when combined (A, B). Killing dynamics of Msm (A) and Mtb (B) (expressing the *luxABCDE* operon from *Photorhabdus luminescens* [*Andreu et al., 2010*]) measured via luciferase production (RLU = relative light units). Amoxicillin (AM) (Msm-1.25; Mtb-3.125 μg/mL); Meropenem (MR) (Msm-10; Mtb-6.25 μg/mL); Amoxicillin + Meropenem: Msm-1.25 μg/mL AM +10 μg/mL MR; Mtb-3.125 μg/mL AM +6.25 μg/mL MR). Biological triplicate are plotted for Mtb. All drugs were used in combination with 5 μg/mL clavulanate.

DOI: https://doi.org/10.7554/eLife.37516.032

The following source data and figure supplements are available for figure 5:

**Source data 1.** Luminescence measurements used for *Figure 5A,B*.
DOI: https://doi.org/10.7554/eLife.37516.036
**Figure supplement 1.** Light production (RLU) correlates to colony-forming units (CFU) in mycobacterial cells expressing *luxABCDE* in drug treatment.
DOI: https://doi.org/10.7554/eLife.37516.033
**Figure supplement 2.** Meropenem and Amoxicillin killing kinetics and minimum inhibitory concentration data.
DOI: https://doi.org/10.7554/eLife.37516.034
**Figure supplement 2—source data 1.** Luminescence measurements used for *Figure 5—figure supplement 2A,B*.
DOI: https://doi.org/10.7554/eLife.37516.035

tetrapeptide substrate for the LDT-dependent exchange reaction. In addition to their ability to exchange D-amino acids onto tetrapeptides, LDTs also catalyze non-canonical 3–3 crosslinks.

Crosslinks catalyzed by LDTs are rare in model rod-shaped bacteria like *E.coli* and *B. subtilis* but, are abundant in polar growing bacteria like mycobacteria, *Agrobacterium tumefaciens* and *Sinorhizobium meliloti* (*Brown et al., 2012*; *Cameron et al., 2015*; *Kumar et al., 2012*; *Lavollay et al., 2008*; *Pisabarro et al., 1985*). Here, we find that Msm cells lacking 3–3 crosslinks cannot maintain rod shape at sites of aging cell wall. 4–3 crosslinks made by PBPs appear able to maintain rod shape near the poles, the sites of newer cell wall (*Figure 6A*). Over time, as older cell wall moves toward the middle of the cell, it loses structural stability, and begins to bleb. The gradual manner in which rod shape is lost in cells lacking 3–3 crosslinks suggests that cell wall processing must occur to destabilize this portion of the rod. Consistent with this idea, we find that an enzyme that cleaves 4–3 crosslinks, the D,D-endopeptidase/D,D-carboxypeptidase DacB2, also localizes to sites of old cell wall and knockdown of this enzyme leads to smaller blebs.

Why would Msm cells create 4–3 crosslinks to eventually cleave them? There are many possibilities. For example, perhaps in the absence of lateral cell wall synthesis, the creation of substrate for LDTs through the destruction of 4–3 crosslinks allows the cell to engage the PG along the lateral cell body. This could be important for altering the thickness of the PG layer or anchoring it to the membrane at sites of otherwise 'inert' cell wall. Additionally, it may be that as PG ages, it is being

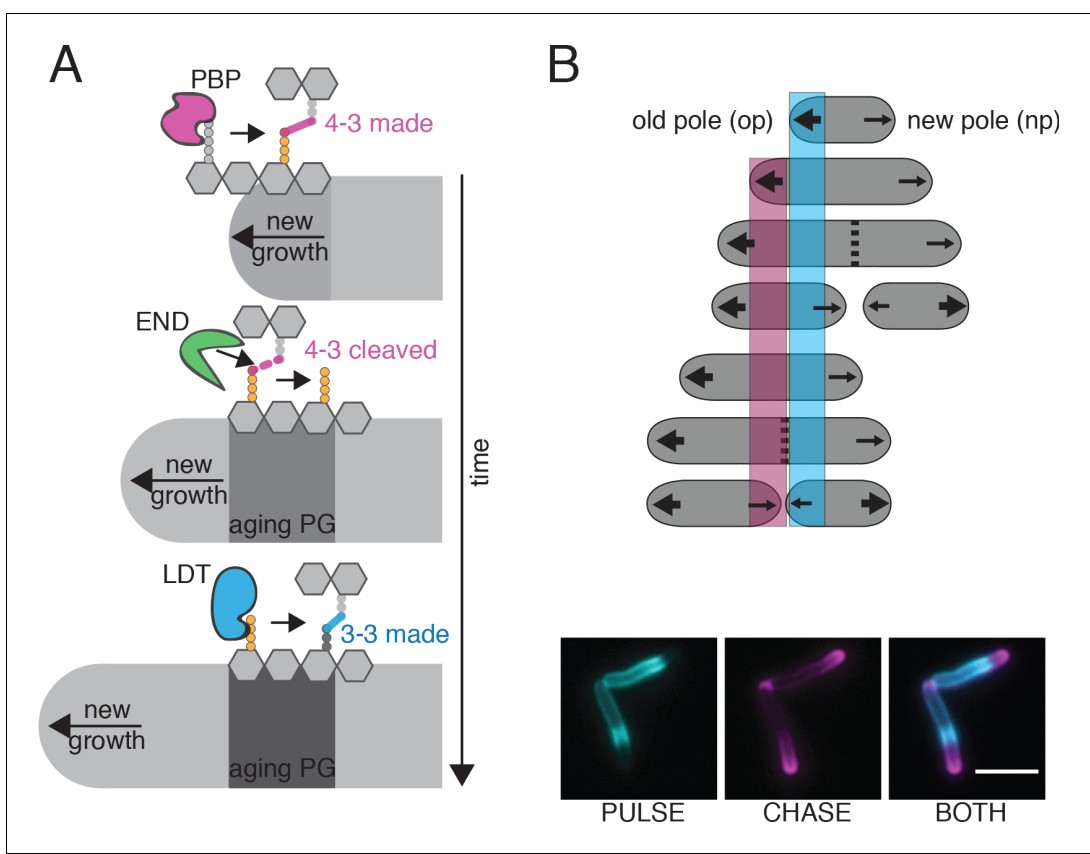

**Figure 6.** Model for PG enzyme and substrate distribution as governed by polar growth and PG segregation by age. (**A**) A model for PG age, PG enzyme and crosslink segregation via polar growth in mycobacteria. First, 4–3 crosslinks are made by PBPs at site of new growth, where the pentapeptide substrate resides. Then, these 4–3 crosslinks can be cleaved by D,D-endopeptidases (END). This action would leave a free tetrapeptide. Lastly, LDTs can utilize this tetrapeptide to generate 3–3 crosslinks. As this is occurring over time and during polar growth, the aging cell wall moves toward mid-cell (new growth at the poles moves away from the existing cell wall). (**B**) Schematic of PG segregation by age (top). 2 min FDAA pulse (cyan), 45 min outgrowth, followed by 2 min FDAA chase (magenta) in WT Msm cells (bottom). Newest cell wall (magenta), older cell wall (cyan). Scale bar = 5 μm.
DOI: https://doi.org/10.7554/eLife.37516.037

manicured or marked for septal synthesis. Supporting this idea, we find that the gradient localization patterns of fluorescently tagged PonA1, LdtE and DacB2, (as well as FDAAs) all have local minima at mid-cell closer to the new pole- a location that correlates with the asymmetric site of division in mycobacteria (*Aldridge et al., 2012*; *Santi et al., 2013*; *Eskandarian et al., 2017*). The lack of localization of PG synthesis enzymes and FDAAs suggests a lack of penta- and tetra- peptide substrates. This implies that this region of the cell may be more abundantly crosslinked, as crosslinking utilizes these peptide species. Could 3–3 crosslinking be a signal for septal placement? Mycobacteria are missing known molecular septal placement mechanisms like the Noc and the Min system (*Hett and Rubin, 2008*). The major septal PG hydrolase is RipA, a D,L-endopeptidase which cleaves the bond between the second and third amino acid of PG side chains, a substrate available on 3–3 crosslinked material (*Böth et al., 2011*; *Vollmer et al., 2008b*). While LdtE-mRFP does not itself strongly localize to the site of division, the crosslinks it synthesizes could migrate toward the mid-cell through polar elongation. Transmitting information from the tip to mid-cell through polar growth was recently described in mycobacteria: atomic force microscopy revealed cell-envelope deformations formed at the pole of Msm travel to mid-cell through polar growth, marking the future site of division (*Eskandarian et al., 2017*). Thus, it is intriguing to speculate that 3–3 crosslinks found at aging cell wall could be important for localizing cell division machinery.

In well-studied rod-shaped bacteria like *E.coli* and *B. subtilis*, shape is maintained by MreB-directed PG synthesis along the lateral cell body (*Garner et al., 2011*; *Hussain et al., 2018*; *Ursell et al., 2014*). On the other hand, mycobacteria maintain shape in the absence of an obvious MreB homolog, and in the absence of lateral cell wall elongation. Furthermore, in contrast to lateral-elongating bacteria, in which new and old cell wall are constantly intermingled during growth, polar growth segregates new and old cell wall (*Figure 6B*). We find that mycobacteria appear to utilize 3–3 crosslinks at asymmetrically distributed aging cell wall to provide stability along the lateral body, something that may not be required in the presence of MreB-directed PG synthesis.

New drug combinations for TB are desperately needed. There has been a renewed interest in repurposing FDA-approved drugs for TB treatment (*Diacon et al., 2016*). Some of that interest has focused on β-lactams, the oldest class of antibiotics which are the therapeutic bedrock for most other infections. We find that the protein targets of two different classes of β-lactams – enzymes which do very similar chemistry – PG crosslinking – are distributed differentially in a single cell and across the population. In the absence of 3–3 crosslinks, 4–3 crosslinks become more important for cell viability. These data predict that a drug combination which inhibits both PBPs and LDTs will work synergistically to more quickly kill Mtb, a prediction we verified *in vitro*. Interestingly, meropenem combined with amoxicillin/clavulanate resulted in early clearance of Mtb from patient sputum (*Diacon et al., 2016*). In fact, the combination might be key to accelerated killing of Mtb (*Gonzalo and Drobniewski, 2013*).

# Materials and methods

**Key resources table**

| Reagent type (species) or resource | Designation | Source or reference | Identifiers | Additional information |
|---|---|---|---|---|
| Strain (*Mycobacterium smegmatis*) | KB85; (WT Msm) | this work | *Mycobacterium smegmatis* mc$^2$155 | Wildtype *M. smegmatis* |
| Strain (*M. smegmatis*) | KB134 | this work | mc2155Δl dtA::loxP | |
| Strain (*M. smegmatis*) | KB156 | this work | mc2155Δl dtA::loxP + ΔldtE:: zeoR | |
| Strain (*M. smegmatis*) | KB200 (ΔldtAEB) | this work | mc2155Δl dtA::loxP ΔldtE:: zeoR + ΔldtB:: hygR | |

*Continued on next page*

Continued

| Reagent type (species) or resource | Designation | Source or reference | Identifiers | Additional information |
|---|---|---|---|---|
| Strain (*M. smegmatis*) | KB209 | this work | mc2155Δ*ldtA*::loxP Δ*ldtE*::loxP Δ*ldtB*::loxP + Δ*ldtC*:: hygR | |
| Strain (*M. smegmatis*) | KB222 | this work | mc2155Δ*ldtA*::loxP Δ*ldtE*::loxP Δ*ldtB*::loxP Δ*ldtC*:: hygR Δ*ldtG*:: zeoR | |
| Strain (*M. smegmatis*) | KB303 (ΔLDT) | this work | mc2155Δ*ldtA*::loxP Δ*ldtE*::loxP Δ*ldtB*::loxP Δ*ldtC*:: loxP Δ*ldtG*:: loxP Δ*ldtF*:: hygR | |
| Strain (*Escherichia coli* XL1-Blue) | KB302 | this work | pTetO-ldtE (MSMEG_0233)- Gly-Gly-Ser linker-mRFP | |
| Strain (*M. smegmatis*) | KB316 (ΔLDTcomp) | this work | [mc2155Δ*ldtA*::loxP Δ*ldtE*::loxP Δ*ldtB*::loxP Δ*ldtC*:: loxP Δ*ldtG*:: loxP Δ*ldtF*:: hygR]+KB302 | |
| Strain (*M. smegmatis*) | KK311 | this work; plasmid from *Kieser et al. (2015a)* | mc2155 + TetO-ponA1-RFP (*Kieser et al., 2015b*) | |
| Strain (*Escherichia coli* Top10) | KB380 | this work | pTetO-dacB2 (MSMEG_2433)- glycine-glycine- serine linker-mRFP | |
| Strain (*M. smegmatis*) | KB414 | this work | mc2155 + KB380 | |
| Strain (*M. smegmatis*) | HR583 | this work | KB303 (ΔLDT)+CRISPRi vector (*Rock et al., 2017*) with *dacB2* targeting sgRNA | Plasmid from Dr. Sarah Fortune (Harvard School of Public Health) and Dr. Jeremy Rock (Rockefeller University) |
| Strain (*E. coli* BL21) | KB428 | this work | *E.coli* BL21 + pET28 b (dacB2) | Plasmid pET28b from Dr. Suzanne Walker |

## Bacterial strains and culture conditions

Unless otherwise stated, *M. smegmatis* (mc$^2$155) was grown shaking at 37°C in liquid 7H9 media consisting of Middlebrook 7H9 salts with 0.2% glycerol, 0.85 g/L NaCl, ADC (5 g/L albumin, 2 g/L dextrose, 0.003 g/L catalase), and 0.05% Tween 80 and plated on LB agar. *M. tuberculosis* (H37Rv) was grown in liquid 7H9 with OADC (oleic acid, albumin, dextrose, catalase) with 0.2% glycerol and 0.05% Tween 80. Antibiotic selection for *M. smegmatis* and *M. tuberculosis* were done at the following concentrations in broth and on agar: 25 µg/mL kanamycin, 50 µg/mL hygromycin, 20 µg/mL zeocin and 20 µg/mL nourseothricin and, twice those concentrations for cloning in *E.coli* (TOP10, XL1-Blue and DH5α).

## Strain construction

### ΔLDT

*M. smegmatis* mc$^2$155 mutants lacking *ldtABECFG* (ΔLDT) was constructed using recombineering to replace endogenous copies with zeocin or hygromycin resistance cassettes flanked by lox sites as previously described (*Boutte et al., 2016*). Briefly, about 500 base pairs of upstream and downstream sequence surrounding the gene of interest were amplified via PCR (KOD Xtreme$^{TM}$ Hot Start DNA polymerase (EMD Millipore, Billerica, MA)). These flanking regions were amplified with overlaps to either a zeocin or hygromycin resistance cassette flanked by loxP sites and these pieces were assembled into a deletion construct via isothermal assembly (*Gibson et al., 2009*). Each deletion cassette was transformed into Msm expressing inducible copies of RecET for recombination (*Murphy et al., 2015*). Once deletions were verified by PCR and sequencing, the antibiotic resistance cassettes were removed by the expression of Cre recombinase. The order of deletion construction in the ΔLDT strain was as follows (where arrows represent transformation of a Cre-recombinase plasmid, followed by curing of the Cre-recombinase plasmid as it contains the *sacB* gene for sucrose counter selection on LB supplemented with 10% sucrose, and strain names are listed in parenthesis). This resulted in the removal of antibiotic cassettes flanked by loxP sites:

1) mc$^2$155Δ*ldtA*:: zeo$^R$ (KB103)→ mc$^2$155Δ*ldtA*::loxP (KB134)

Sequence flanking *ldtA* upstream was amplified with KB208/209; downstream flanking sequence was amplified with KB210/211

2) mc$^2$155Δ*ldtA::loxP* +Δ*ldtE*:: zeo$^R$ (KB156)

Sequence flanking *ldtE* upstream was amplified with KB220/221; downstream flanking sequence was amplified with KB222/223

3) mc$^2$155Δ*ldtA::loxP* Δ*ldtE*:: zeo$^R$ + Δ*ldtB*:: hyg$^R$ (KB200) → mc$^2$155Δ*ldtA::loxP* Δ*ldtE*::loxP Δ*ldtB*::loxP (KB207)

Sequence flanking *ldtB* upstream was amplified with KB444/445; downstream flanking sequence was amplified with KB446/447

4) mc$^2$155Δ*ldtA::loxP*Δ*ldtE*::loxP Δ*ldtB*::loxP + Δ*ldtC*:: hyg$^R$ (KB209)

Sequence flanking *ldtC* upstream was amplified with KB216/448; downstream flanking sequence was amplified with KB449/219

5) mc$^2$155Δ*ldtA::loxP* Δ*ldtE*::loxP Δ*ldtB*::loxP Δ*ldtC*:: hyg$^R$ Δ*ldtG*:: zeo$^R$ (KB222)→ mc$^2$155Δ*ldtA*::loxP Δ*ldtE*::loxP Δ*ldtB*::loxP Δ*ldtC*:: loxP Δ*ldtG*:: loxP (KB241)

Sequence flanking *ldtG* upstream was amplified with KB228/454; downstream flanking sequence was amplified with KB455/231

6) mc$^2$155Δ*ldtA*::loxP Δ*ldtE*::loxP Δ*ldtB*::loxP Δ*ldtC*:: loxP Δ*ldtG*:: loxP Δ*ldtF*:: hyg$^R$

Sequence flanking *ldtF* upstream was amplified with KB224/452; downstream flanking sequence was amplified with KB453/227

(KB303 referred to as ΔLDT).

### Mtb and Msm Lux

*M. tuberculosis* H37Rv was transformed with a vector expressing the codon optimized *Photorhabdus luminescens luxABCDE* operon (pMV306hsp + LuxG13 –Addgene #26161; RRID:SCR_005907) (*Andreu et al., 2010*). This strain is referred to as Mtb Lux. The same plasmid was transformed into Msm and this strain is referred to as Msm Lux.

### ΔLDT$_{comp}$

To complement ΔLDT (KB303) we placed a copy of *ldtE* (*MSMEG_0233*) under the constitutive TetO promoter (a UV15 derivative within a pMC1s plasmid that is inducible with anhydro-tetracycline in the presence of a tet-repressor TetR, which the ΔLDT$_{comp}$ strain lacks [*Kieser et al., 2015b*]) on vector that integrates at the L5 phage integration site of the chromosome of the ΔLDT strain (the vector is marked with kanamycin resistance). A glycine, glycine, serine linker was cloned between *ldtE* and mRFP in this complementation construct. LdtE lacking a stop codon, with a glycine-glycine-serine linker was amplified with primers 323A/351. The fluorescent protein mRFP was amplified with primers KB352/353 with overlaps to the linker and the vector backbone.

### DacB2 expression strain

A truncated MSMEG_2433 was cloned into pET28b for isopropyl β-D-1-thiogalactopyranoside (IPTG) inducible expression in *E. coli* BL21 (DE3). MSMEG_2433$_{(29-296)}$ amplified using the primers KB662/663 with overlaps to NdeI digested pET28b. This PCR product was assembled with the digested vector using isothermal assembly. The resulting vector was transformed into *E. coli* BL21.

### PonA1-RFP

PonA1-RFP was amplified from *Kieser et al. (2015a)* using primers KK1/KK2 and KK3/KK4, digested with NdeI and cloned into the same vector as ldtE-mRFP (see above).

### DacB2-mRFP

Using the same cloning strategy as for LdtE-mRFP, *dacB2* (MSMEG_2433) was amplified lacking a stop codon, with a gly-gly-ser linker using primers KB626/627. The fluorescent protein mRFP was amplified with primers KB628/353 containing overlaps to *dacB2*, the linker and the vector backbone. The resulting vector was transformed into WT Msm.

### PonA1 transpeptidase essentiality L5 allele swapping

To test essentiality of transpeptidation by PonA1 in the ΔLDT cells, L5 allele swapping as described in *Kieser et al., 2015b* was performed. The plasmids used in this experiment were previously published in *Kieser et al., 2015b*. Briefly, a wild-type copy of PonA1 (TetO driven expression, L5 integrating and kanamycin marked) was transformed into ΔLDT. Then, the endogenous copy of *ponA1* was replaced with zeocin using the above mentioned recombineering technique (amplifying the construct from a previously published deletion mutation of *ponA1* [*Kieser et al., 2015b*]). Swapping efficiency of either wildtype or transpeptidase mutant PonA1 marked with nourseothricin was tested with a transformation into ΔLDT//L5-TetO-ponA1 (WT)-kanamycin.

## Whole genome sequencing

Whole genome sequencing was performed on wild-type mc$^2$155 as well as the ΔLDT mutant. Sequencing was done on an Illumina HiSeq 4000 (RRID:SCR_016386) with 150 bp paired-end reads. There was a mean depth of coverage of 148x. All 6 LDT genes were verified as deleted. Furthermore, there was no evidence of any duplications or cross-over events based on a coverage plot.

The sequencing has been uploaded to NCBI's SRA (details for sample identifiers are provided below).

STUDY: PRJNA451029 (SRP141343)
ΔLDT SAMPLE: deltaLdtAEBCGF (SRS3442031)
ΔLDT EXPERIMENT: deltaLdtAEBCGF (SRX4275943)
ΔLDT RUN: deltaLdtAEBCGF_R2.fastq (SRR7403831)
WT SAMPLE: Msmeg-KB (SRS3442032)
WT EXPERIMENT: Msmeg-KB (SRX4275944)
WT RUN: Msmeg-KB_R2.fastq (SRR7403830)

## *M. tuberculosis* and *M. smegmatis* minimum inhibitory concentration (MIC) determination

Mtb or Msm Lux was grown to log phase and diluted to an OD$_{600}$ = 0.006 in each well of non-treated 96-well plates (Genesee Scientific) containing 100 µL of meropenem (Sigma Aldrich) and/or amoxicillin (Sigma Aldrich) diluted in 7H9 + OADC + 5 µg/mL clavulanate (Sigma Aldrich). Msm media contained ADC rather than OADC. Cells were incubated in drug at 37°C shaking for 7 days (Mtb) or 1 day (Msm), 0.002% resazurin (Sigma Aldrich) was added to each well, and the plates were incubated for 24 hr before MICs were determined. Pink wells signify metabolic activity and blue signify no metabolic activity. (*Kieser et al., 2015a*) Checkerboard MIC plates and fractional inhibitory concentrations were calculated as described in (*Synergism Testing: Broth Microdilution Checkerboard and Broth Macrodilution Methods, 2016*).

### *M. tuberculosis* and *M. smegmatis* drug killing assays

Mtb Lux was grown to log phase (kanamycin 25 µg/mL) and diluted in 30 mL inkwells (Corning Lifesciences) to an $OD_{600}$ = 0.05 in 7H9 + OADC + 5 µg/mL clavulanate with varying concentrations of amoxicillin, meropenem, or both. 100 µL of these cultures were pipetted in duplicate into a white 96-well polystyrene plate (Greiner Bio-One) and luminescence was measured in a Synergy H1 microplate reader from BioTek Instruments, Inc. using the Gen5 Software (2.02.11 Installation version). The correlation between relative light units (RLU) and CFU is shown in Msm in *Figure 5—figure supplement 1*.

Msm Lux was grown to log phase and diluted into white 96-well polystyrene plates to an $OD_{600}$ = 0.05. Plates were sealed with 4titude Moisture Barrier Seals and shaken continuously at 37° C. Luminescence measurements (RLU) were taken at 15-min intervals integrated over 1000 ms in a TECAN Spark 10M plate reader for 18 hr.

### Fluorescent D-amino acid labeling

NADA (3-[7-nitrobenzofurazan]-carboxamide-D-Alanine), HADA (3-[7-hydroxycoumarin]-carboxamide-D-Alanine) and TADA (3-[5-carboxytetramethylrhodamine]-carboxamide-D-Alanine) were synthesized by Tocris following the published protocol (*Kuru et al., 2015*). To 1 mL of exponentially growing cells 0.1 mM of FDAA final was added and incubated for 2 min before washing in 7H9 twice. For still imaging, after the second wash, cells were fixed in 7H9 + 1% paraformaldehyde before imaging. For pulse chase experiments, cells were stained, washed with 7H9 and allowed to grow out for 40 min before being stained with a second dye and imaged.

### Flow cytometry

An *M. smegmatis* transposon library was grown to mid-log phase, and stained with 2 µg/mL NADA for 2 min. Cells were centrifuged and half of the supernatant was discarded. The pellet was resuspended in the remaining supernatant, passed through a 10 µm filter and taken to be sorted (FACSAria; Excitation: 488 nm; Emission filter: 530/30; RRID:SCR_009839). Two bins were drawn at the lowest and highest staining end of the population, representing 12.5% of the population. 600,000 cells from these bins were sorted into 7H9 medium. Half of this was directly plated onto LB agar supplemented with kanamycin to select for cells harboring the transposon. The remaining 300,000 cells were grown out in 7H9 to log phase, stained with FDAA and sorted again to enrich the populations.

### Transposon sequencing, mapping and FDAA flow cytometry enrichment analysis

Genomic DNA (gDNA) was harvested from the sorted transposon library colonies and transposon-gDNA junction libraries were constructed and sequenced using the Illumina Hi-Seq platform (*Long et al., 2015*). Reads were mapped on the *M. smegmatis* genome, tallied and reads at each TA site for the bins (low/high incorporating sort 1 and 2) were imported into MATLAB and processed by a custom scripts as described in *Rego et al. (2017)*. Source code for this analysis can be found on GitHub at: https://github.com/hesperrego/baranowski_2018 (copy archived at https://github.com/elifesciences-publications/baranowski_2018).

Sequencing data are available in NCBI's SRA with accession number SRP141343.

### Microscopy

Both still imaging and time-lapse microscopy were performed on an inverted Nikon TI-E microscope at 60x magnification. Time-lapse was done using a CellASIC ONIX2 Microfluidic System (Millipore Sigma, B04A plate) with constant liquid 7H9 flow in a 37°C chamber. For turgor experiment (*Figure 2A*), cells were grown in either 7H9 or 7H9 500 mM sorbitol overnight, and then switched to either 7H9 with 150 mM sorbitol (high osmolar) or to 7H9 alone (iso-osmolar).

### Atomic force microscopy

AFM experimentation was conducted as previously(*Eskandarian et al., 2017*). In short, polydimethylsiloxane (PDMS) – coated coverslips were prepared by spin-coating a mixture of PDMS at a ratio of 15:1 (elastomer:curing agent) with hexane (Sigma 296090) at a ratio of 1:10 (PDMS:hexane) (*Koschwanez et al., 2009*; *Thangawng et al., 2007*). A 50 µl filtered (0.5 µm pore size PVDF filter –

Millipore) aliquot of bacteria grown to mid-exponential phase and concentrated from 2 to 5 ml of culture was deposited onto the hydrophobic surface of a PDMS-coated coverslip and incubated for ~20 min to increase surface interactions between bacteria and the coverslip. 7H9 medium (~3 ml) was supplied to the sample so as to immerse the bacterial sample and the AFM cantilever in fluid. The AFM imaging mode, Peak Force QNM, was used to image bacteria with a Nanoscope five controller (Veeco Metrology) at a scan rate of 0.5 Hz and a maximum Z-range of 12 μm. A ScanAsyst fluid cantilever (Bruker) was used. Height, peak force error, DMT modulus, and log DMT modulus were recorded for all scanned images in the trace and retrace directions. Images were processed using Gwyddion (Department of Nanometrology, Czech Metrology Institute). ImageJ was used for extracting bacterial cell profiles in a tabular form.

## Correlated optical fluorescence and AFM

Correlated optical fluorescence and AFM images were acquired as described (*Eskandarian et al., 2017*). Briefly, optical fluorescence images were acquired with an electron-multiplying charge-coupled device (EMCCD) iXon Ultra 897 camera (Andor) mounted on an IX81 inverted optical microscope (Olympus) equipped with an UPLFLN100XO2PH x100 oil immersion objective (Olympus). Transmitted light illumination was provided by a 12V/100W AHS-LAMP halogen lamp. An U-MGFPHQ fluorescence filter cube for GFP with HQ-Ion-coated filters was used to detect GFP fluorescence. The AFM was mounted on top of the inverted microscope, and images were acquired with a Dimension Icon scan head (Bruker) using ScanAsyst fluid cantilevers (Bruker) with a nominal spring constant of 0.7 N m$^{-1}$ in Peak Force QNM mode at a force setpoint ~1 nN and typical scan rates of 0.5 Hz. Indentation on the cell surface was estimated to be ~10 nm in the Z-axis. Optical fluorescence microscopy was used to identify Wag31-GFP puncta expressed in a wild-type background (*Santi et al., 2013*) in order to distinguish them from cells of the ΔLDT mutant strains.

## Calculating cell surface rigidity

A cell profile was extracted from AFM Height and DMT Modulus image channels as sequentially connected linear segments following the midline of an individual cell. A background correction was conducted to by dividing the DMT modulus values of the cell surface by the mean value of the PDMS surface and rescaled to compare the cell surface rigidity between individual cells from different experiments. The DMT modulus reflects the elastic modulus (stress-strain relationship) for each cross-sectional increment along the cell length.

## Analysis of fluorescent protein distribution

Using a segmented line, profiles of cells from new to old pole were created at the frame 'pre-division' based on physical cell separation of the phase image. A custom FIJI (*Schindelin et al., 2012*) script was run to extract fluorescence line profiles of each cell and save them as. csv files. These. csv files were imported to Matlab where a custom script was applied to normalize the fluorescence line profile to fractional cell length and to interpolate the fluorescence values to allow for averaging. Source code for this analysis can be found on GitHub at:https://github.com/hesperrego/baranowski_2018

## Analysis of cell wall distribution

Cells were stained with AlexaFluor 488 NHS ester (ThermoFisher Scientific) as described previously (*Aldridge et al., 2012*) and followed via time-lapse microscopy in the CellASIC device. Briefly, 1 mL of log phase cells was pelleted at 8000 rpm for 1 min and washed with 1 mL PBST. The pellet was resuspended in 100 uL of PBST and 10 uL Alexa Fluor 488 carboxylic acid succinimidyl ester was added for a final concentration of 0.05 mg/mL. This was incubated for 3 min at room temperature. Stained cells were pelleted for 1 min at 13,000 rpm and washed with 500 μL PBST. They were spun again and resuspended in 7H9 for outgrowth observation over time in the CellASIC device.

## Analysis of FDAAs

Images were analyzed using a combination of Oufti (*Paintdakhi et al., 2016*) (RRID:SCR_016244) for cell selection followed by custom coded Matlab scripts to plot FDAA fluorescence over normalized cell length, calculate cell length and bin cells by existence of an FDAA labeled septum. This code

and a manual for its use has been included as a source code file with this manuscript (Source Code-Instructions and code for FDAA image analysis in *Figure 1*).

## Generation of transposon libraries

*M. smegmatis* cells were transduced at ($OD_{600}$ 1.1–1.7) with $\varphi$MycoMarT7 phage (temperature sensitive) that has a Kanamycin marked Mariner transposon as previously described (*Long et al., 2015*). Briefly, mutagenized cells were plated at 37°C on LB plates supplemented with Kanamycin to select for phage transduced cells. Roughly 100,000 colonies per library were scraped, and genomic DNA was extracted. Sequencing libraries were generated specifically containing transposon disrupted DNA. Libraries were sequenced on the Illumina platform. Data were analyzed using the TRANSIT pipeline (*DeJesus et al., 2015*) (RRID:SCR_016492).

Sequencing data are available in NCBI's SRA with accession number SRP141343.

## Peptidoglycan isolation and analysis

600 mL of wild-type and ΔLDT cells were grown to log phase and collected via centrifugation at 5000 x g for 10 min at 4°C. The resulting pellet was resuspended in PBS and cells were lysed using a cell disruptor at 35,000 psi twice. Lysed cells were boiled in 10% SDS (sodium dodecyl sulfate) for 30 min and peptidoglycan was collected via centrifugation at 17,000 x g. Pellets were washed with 0.01% DDM(*n*-Dodecyl β-D-maltoside) to remove SDS and resuspended in 1XPBS + 0.01% DDM. PG was digested with alpha amylase (Sigma A-6380) and alpha chymotrypsin (Amersco 0164) overnight. The samples were again boiled in 10% SDS and washed in 0.01% DDM. The resulting pellet was resuspended in 400 μL 25 mM sodium phosphate pH6, 0.5 mM MgCl2, 0.01% DDM. 20 μL of lysozyme (10 mg/mL) and 20 μL 5 U/μL mutanolysin (Sigma M9901) were added and incubated overnight at 37°C. Samples were heated at 100°C and centrifuged at 100,000 x g. 128 μL of ammonium hydroxide was added and incubated for 5 hr at 37°C. This reaction was neutralized with 122 μL of glacial acetic acid. Samples were lyophilized, resuspended in 300 μL 0.1% formic acid and subjected to analysis by LC-MS/MS. Peptide fragments were separated with an Agilent Technologies 1200 series HPLC on a Nucleosil C18 column (5 μm 100A 4.6 × 250 mm) at 0.5 mL/min flow rate with the following method: Buffer A = 0.1% Formic Acid; Buffer B = 0.1% Formic Acid in acetonitrile; 0% B from 0 to 10 min, 0–20% B from 10 to 100 min, 20% B from 100 to 120 min, 20–80% B from 120 to 130 min, 80% B from 130 to 140 min, 80–0% B from 140 to 150 min, 0% B from 150 to 170 min. MS/MS was conducted in positive ion mode using electrospray ionization on an Agilent Q-TOF (6520).

## Expression and purification of MSMEG_2433 (DacB2)

MSMEG_2433 was expressed and purified using a modified method for purification of low-molecular-weight PBPs that was previously published (*Qiao et al., 2014*). An N-terminally truncated MSMEG_2433$_{(29-296)}$ was cloned into the pET28b vector for isopropyl β-D-1-thiogalactopyranoside (IPTG) inducible expression in *E. coli* BL21 (DE3) (see strain construction notes above). 10mLs of overnight culture grown in LB with Kanamycin (50 μg/mL) were diluted 1:100 into 1 L of LB with Kanamycin (50 μg/mL) and grown at 37°C until an $OD_{600}$ of 0.5. The culture was cooled to room temperature, induced with 0.5 mM IPTG, and shaken at 16°C overnight. Cells were pelleted via centrifugation at 4000 rpm for 20 min at 4°C. The pellet was suspended in 20 mL binding buffer (20 mM Tris pH 8, 10 mM MgCl$_2$, 160 mM NaCl, 20 mM imidazole) with 1 mM phenylmethylsulfonylfluoride (PMSF) and 500 μg/mL DNase. Cells were lysed via three passage through a cell disrupter at ≥10,000 psi. Lysate was pelleted by ultracentrifugation (90,000 × g, 30 min, 4°C). To the supernatant, 1.0 mL washed Ni-NTA resin (Qiagen) was added and the mixture rocked at 4°C for 40 min. After loading onto a gravity column, the resin was washed twice with 10 mL wash buffer (20 mM Tris pH 8, 500 mM NaCl, 20 mM imidazole, 0.1% Triton X-100). The protein was eluted in 10 mL of elution buffer (20 mM Tris pH8, 150 mM NaCl, 300 mM imidazole, 0.1% reduced Triton X-100) and was concentrated to 1 mL with a 10kD MWCO Amicon Ultra Centrifuge Filter. The final protein concentration was measured by reading absorbance at 280 nm and using the estimated extinction coefficient (29459 $M^{-1}cm^{-1}$) calculate concentration. The protein was diluted to 200 μM in elution buffer with 10% glycerol, aliquoted, and stored at −80°C.

Proper folding of purified MSMEG_2433$_{(29-296)}$ was tested via Bocillin-FL binding. Briefly, 20 μM of purified protein was added to penicillin G (100, 1000 U/mL in 20 mM K$_2$HPO$_4$, 140 mM NaCl,

pH7.5) in a 9 µL reaction. The reaction was incubated at 37°C for 1 hr. 10 µM Bocillin-FL was added and incubated at 37°C for 30 min. SDS loading dye was added the quench the reaction and samples were loaded onto a 4–20% gel. MSMEG_2433$_{(29-296)}$ bound by Bocillin-FL was imaged using a Typhoon 9400 Variable Mode Imager (GE Healthcare) (Alexa Excitation-488nm Emission-526nm).

### Lipid II extraction

*B. subtilis* Lipid II was extracted as previously published (*Qiao et al., 2017*).

### SgtB purification

*S. aureus* SgtB was purified as previously published (*Rebets et al., 2014*).

### Purification of *B. subtilis* PBP1

Purification of *B. subtilis* PBP1 was carried out as previously described (*Lebar et al., 2014*).

### *In vitro* Lipid II polymerization and crosslinking

20 µM purified BS Lipid II was incubated in reaction buffer (50 mM HEPES pH 7.5, 10 mM CaCl$_2$) with either 5 µM PBP1 or 0.33 µM SgtB for 1 hr at room temperature. The enzymes were heat denatured at 95°C for 5 min. Purified MSMEG_2433$_{(29-296)}$ was added (20 uM, final) and the reaction was incubated at room temperature for 1 hr. Mutanolysin (1 µL of a 4000 U/mL stock) was added and incubated for 1.5 hr at 37°C (twice). The resulting muropeptides were reduced with 30 µL of NaBH$_4$ (10 mg/mL) for 20 min at room temperature with tube flicking every 5 min to mix. The pH was adjusted to ~4 using with 20% H$_3$PO$_4$ and the resulting product was lyophilized to dryness. The residue was resuspended in 18 µL of water and analyzed via LC-MS as previously reported (*Welsh et al., 2017*).

### Experimental replicates

Biological replicates – independent cultures; Technical replicates – the same culture in replicate.

Microscopy for *Figure 1E,F* and *Figure 1—figure supplement 2* was done once and analyzed. The data shown in *Figure 1—figure supplement 3* was done once in technical triplicate. The graph shows one replicate.

Time-lapse experiment in *Figure 2A* was done twice (biological duplicate on separate days). Included for this figure are videos of full fields of view of the time-lapse experiments (*Figure 2—video 1*- full field; *Figure 2—video 2*- full field). Microscopy for *Figure 2B* was done in biological triplicate on three separate days. The time-lapse phenotype highlighted *Figure 2—figure supplement 1* was observed in biological triplicate on 3 independent days. AFM data in *Figure 2D and E* was derived from two independent experiments done on separate days.

Allele swapping experiment in *Figure 3C* was done once.

Time-lapse microscopy for *Figure 4A* was performed in biological duplicate. The graph in *Figure 4B* represents data from one experiment. *Figure 4C* is representative data from two technical replicates (the same protein and substrate preparations were used). Microscopy and quantification of bleb size in *dacB2* CRISPRi knock-down (*Figure 4—figure supplement 3*) was done twice (biological duplicate on separate days).

Luciferase Msm data in *Figure 5A* was performed once. Luciferase Mtb survival data in *Figure 5B* was done in biological triplicate and technical triplicate. Biological triplicates are plotted. Minimum inhibitory concentrations (MIC) were determined in biological duplicate (two separate cultures on two separate days) and technical duplicate for *Figure 5B*. Combination MIC for *Figure 5—figure supplement 2* was determined once for Mtb and twice for Msm strains.

Fluorescent D-amino acid pulse-chase for *Figure 6B* was done on two independent days (biological duplicate).

## Acknowledgements

We thank Cara Boutte and Erkin Kuru for discussion and thorough manuscript review. We are grateful to the Rubin and Fortune laboratory for discussion and input. We also thank the Microbiology and Immunobiology department at Harvard Medical School for sharing equipment and reagents, as

well as the BSL3 staff at the Harvard School of Public health for tremendous support. This work was supported by the National Institutes of Health U19 AI107774 to EJR and TRI, F32AI104287 to EHR (who is also supported by a Career Award at the Scientific Interface from BWF), R01 GM76710 to SW, R01AI083365 and U19AI109764 to TGB. MAW is supported by an F32 GM123579. LTS was supported by an American Heart Association Postdoctoral fellowship (14POST18480014). HCL is funded by a Simons Foundation Fellow of the Life Sciences Research Foundation award. This work was also supported in part by the Swiss National Science Foundation (310030_156945) and the Innovative Medicines Initiative (115337) to JDM, and from the Swiss National Science Foundation (205321_134786 and 205320_152675), the European Union FP7/2007-2013/ERC Grant Agreement No. 307338 (NaMic), and EU-FP7/Eurostars E!8213 to GEF Support for HAE comes from a European Molecular Biology Organization Long Term Fellowships (191–2014 and 750–2016). KJK was supported by the National Science Foundation Graduate Research Fellowship (DGE1144152, DGE0946799).

# Additional information

## Funding

| Funder | Grant reference number | Author |
|---|---|---|
| National Institutes of Health | U19 AI107774 | Thomas R Ioerger<br>Eric J Rubin |
| Burroughs Wellcome Fund | Career Award at the Scientific Interface | E Hesper Rego |
| American Heart Association | 14POST18480014 | Lok-To Sham |
| Simons Foundation | Fellow of the Life Sciences Research Foundation Award | Hoong C Lim |
| Schweizerischer Nationalfonds zur Förderung der Wissenschaftlichen Forschung | 310030_156945 | John D McKinney |
| Seventh Framework Programme | FP7/2007-2013/ERC Grant agreement No. 307338 (NaMic) | Georg E Fantner |
| European Molecular Biology Organization | 191-2014 | Haig A Eskandarian |
| National Science Foundation | DGE1144152 | Karen J Kieser |
| National Institutes of Health | F32AI104287 | E Hesper Rego |
| National Institutes of Health | R01 GM76710 | Suzanne Walker |
| National Institutes of Health | R01AI083365 | Thomas G Bernhardt |
| National Institutes of Health | F32GM123579 | Michael A Welsh |
| Schweizerischer Nationalfonds zur Förderung der Wissenschaftlichen Forschung | 205321_134786 | Georg E Fantner |
| Innovative Medicines Initiative | 115337 | John D McKinney |
| EU-FP7/Eurostars | E!8213 | Georg E Fantner |
| European Molecular Biology Organization | 750-2016 | Haig A Eskandarian |
| National Science Foundation | DGE0946799 | Karen J Kieser |
| National Institutes of Health | U19AI109764 | Thomas G Bernhardt |
| Schweizerischer Nationalfonds zur Förderung der Wissenschaftlichen Forschung | 205320_152675 | Georg E Fantner |

The funders had no role in study design, data collection and interpretation, or the decision to submit the work for publication.

## Author contributions
Catherine Baranowski, Conceptualization, Formal analysis, Validation, Investigation, Visualization, Methodology, Writing—original draft; Michael A Welsh, Investigation, Visualization, Methodology, Writing—review and editing; Lok-To Sham, Formal analysis, Investigation, Methodology; Haig A Eskandarian, Formal analysis, Investigation, Methodology, Writing—review and editing; Hoong Chuin Lim, Software, Formal analysis, Visualization, Methodology; Karen J Kieser, Resources, Writing—review and editing; Jeffrey C Wagner, Investigation; John D McKinney, Georg E Fantner, Suzanne Walker, Resources, Supervision, Funding acquisition; Thomas R Ioerger, Data curation, Formal analysis, Writing—review and editing; Thomas G Bernhardt, Resources, Supervision, Funding acquisition, Writing—review and editing; Eric J Rubin, Conceptualization, Resources, Supervision, Funding acquisition, Writing—review and editing; E Hesper Rego, Conceptualization, Resources, Software, Formal analysis, Supervision, Investigation, Writing—review and editing

## Author ORCIDs
Catherine Baranowski (iD) https://orcid.org/0000-0003-0407-8609
Michael A Welsh (iD) http://orcid.org/0000-0001-8268-6285
Haig A Eskandarian (iD) http://orcid.org/0000-0002-0610-0550
Thomas G Bernhardt (iD) http://orcid.org/0000-0003-3566-7756
Eric J Rubin (iD) https://orcid.org/0000-0001-5120-962X
E Hesper Rego (iD) http://orcid.org/0000-0002-2973-8354

## Decision letter and Author response
Decision letter https://doi.org/10.7554/eLife.37516.041
Author response https://doi.org/10.7554/eLife.37516.042

# Additional files

## Supplementary files
• Supplementary file 1. List of primers.
DOI: https://doi.org/10.7554/eLife.37516.038

## Data availability
Sequencing data were deposited into NCBI's Sequence Read Archive (SRA) under SRA study-SRP141343 https://www.ncbi.nlm.nih.gov/Traces/study/?acc=SRP141343.

The following dataset was generated:

| Author(s) | Year | Dataset title | Dataset URL | Database and Identifier |
|---|---|---|---|---|
| Ioerger T, Baranowski C, Rubin EJ, Rego EH | 2018 | Mycobacterium smegmatis LDT mutant Tnseq, LDT mutant and WT WGS, FDAA FACS Tnseq screen | https://www.ncbi.nlm.nih.gov/Traces/study/?acc=SRP141343 | NCBI Sequence Read Archive, SRP141343 |

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
