## [Decision Letter]

Thank you for submitting your article "Maturing Mycobacterial Peptidoglycan Requires Non-canonical Crosslinks to Maintain Shape" for consideration by *eLife*. Your article has been reviewed by four peer reviewers, one of whom is a member of our Board of Reviewing Editors, and the evaluation has been overseen by Wendy Garrett as the Senior Editor. The following individuals involved in review of your submission have agreed to reveal their identity: Michel Arthur (Reviewer #2); Clif Barry (Reviewer #3); Gyanu Lamichhane (Reviewer #4).

The reviewers have discussed the reviews with one another and the Reviewing Editor has drafted this decision to help you prepare a revised submission.

Summary:

The manuscript by Dr. Rego and colleagues explores peptidoglycan (PG) metabolism in mycobacteria. The biosynthetic pathway for the PG polymer in these organisms represents several departures from well-known paradigms, most notably; mycobacteria extend their cell surfaces by incorporating new material at the cell tips. Also, the composition of cross-linked mycobacterial PG comprises a large proportion of 3-3 crosslinks, which are relatively rare in model organisms wherein PG has been intensely studied. This form of cross linking is mediated by the activity of L,D-transpeptidases (LDTs) and the authors focus their efforts on describing the function of these enzymes in *Mycobacterium smegmatis*. Using fluorescent D-amino acids (FDAAs) to visualise PG synthesis, they report strong staining of an FDAA at the cell poles, with a gradient of staining that extends to mid-cell, from the old pole. To determine which enzymes drive this pattern of staining the authors carry out saturating transposon mutagenesis, together with fluorescence associated cell sorting, to identify cells where staining was reduced. This analysis identified three LDTs (LDTA, B, and E) and combinatorial deletion of these resulted in reduced FDAA uptake. The authors proceed to delete all six LDTs (ΔLDT) and note similar effects, indicating that FDAA incorporation in mycobacteria is LDT-dependent. Next, the authors characterise the morphological defects that occur upon combinatorial deletion of all LDTs and report that a proportion of cells lose rod shape, which could be rescued by high osmolarity, suggesting that loss of cell wall rigidity mediated shape defects, a notion supported by measurement of cell rigidity using atomic force microscopy. Time-lapse microscopy revealed that the daughter cell that lost rod shape inherited the new pole and oldest cell wall, with spherical blebs localized in the cell retaining the old cell wall. This suggested that LDTs stabilize old cell wall material. After this, the authors demonstrate that in the absence of LDTs, mycobacteria depend on PbpA and PonA2-mediated 4-3 crosslink formation to maintain cell wall integrity. The authors then demonstrate that PG synthesizing and remodelling enzymes locate to distinct sites of the cell. Combinatorial inhibition of both 4-3 and 3-3 crosslink formation by antibiotics illustrated the potential tractability of targeting PG for tuberculosis drug treatment.

Central conclusions:

1) LDT-mediated 3-3 crosslinks are required for maintenance of rod shape in *M. smegmatis*.

2) The different types of cross-links, 4-3 versus 3-3, are most likely spatially distributed to distinct locations in the mycobacterial cell, with mature PG locating to the sidewall in a 3-3 crosslinked conformation.

3) In the absence of the ability to generate 3-3 crosslinks, mycobacteria become critically reliant of 4-3 crosslinks generated by HMW PBPs.

The study is carefully conducted and opens new avenues to decipher the mechanism underlying septum localization in mycobacteria and to identify strategies for synergistic killing of mycobacteria by drug combinations.

Essential revisions:

1) In Figure 1B, how do the authors determine the new from the old pole? This most likely related to birth size and intensity of staining at the cell pole, but some illustration of this would be useful in the figure and is required to understand the rest of the work.

2) Data demonstrating how the genotype of the Msm strains lacking all 6 (and 3) LDT-encoding genes was verified is necessary. It could be Southern blotting or sequencing. Sequencing is mentioned in the Materials and methods section, but it is not clear what was done to ensure that there were no unintended cross-over events elsewhere in the chromosome as well.

3) Why just complement with LDTE? Were the other three LDTs enriched/identified in the screen? Was the sextuple mutant necessary to completely eliminate FDAA staining? In this context, for rigour, genetic complementation of the LDT triple mutant is necessary, this seems relatively easy. The same construct used to complement the ΔLDT mutant can be used in this case.

4) The study unambiguously shows that DacB2 has endopeptidase activity (cleavage of 4-3 crosslinks) and that DacB2 co-localizes with LDTE. The authors conclude that the endopeptidase activity of DacB2 is responsible for the formation of blebs in the absence of LDTs. Direct evidence of the proposed model should be obtained by constructing a *dacB2* ΔLDT mutant. As *dacB2* was deemed to be nonessential in genome wide inactivation studies in *M. tuberculosis* and direct gene deletion (Microb Pathog. 2012 52:109-16), this experiment should be feasible and is required to support the conclusions of the study. Genetic complementation of the resulting combinatorial mutant (*dacB2* ΔLDT mutant) is also required.

5) The MS analysis of peptidoglycan structure appearing in "Figure 1—figure supplement 3B" raises numerous issues including the absence of any definition of the molecular ions that were detected and the inversion of the donor and acceptor stems between the sets of 4-3 and 3-3 crosslinked dimers. There are inconsistencies between the observed mass and the proposed structures, e.g. peaks 7 and 14 display the same mass but contain 2 and 4 NH2, respectively. Please, be consistent and list the number of NH2 rather than a mixed designation with OH for certain peaks. More importantly, peak 10 disappeared upon deletion of the LDT genes although it is reported to contain a 4-3 crosslink (formed by PPBs). These shortfalls must be addressed.

6) The experiments showing the cellular localization of the various proteins are not really as clear. Figure 4, subsection “Peptidoglycan synthesizing enzymes localize to differentially aged cell wall”, first paragraph: While it is clear that LDTE-mRFP localized farther from the poles, it appears PonA1-RFP localizes throughout the cell wall as the staining is clearly visible throughout the cell.

Subsection “Peptidoglycan synthesizing enzymes localize to differentially aged cell wall”, last paragraph: Figure 4A, bottom panel, shows prominent staining in the poles as well. Therefore, the conclusion that 'DacB2-mRFP localized closer to LDT-mRFP' is not entirely supported by the data.

Similarly, from the Discussion (end of third paragraph) to conclude that 3-3 cross-linkages are responsible for localizing the biosynthetic machinery seems inadequate from the location of only one of these proteins. These shortfalls can be addressed by attempting localization of another LDT homologue and providing clearer pictures of how PonA1-RFP localizes. Please ensure that any quantification of localization intensity is done for a sufficient number of cells to ensure robustness.

7) Figure 6A, subsection “Mycobacteria are hypersensitive to PBP inactivation in the absence of LDTs”, first paragraph: It is proposed that D,D-endopeptidase generates tetrapeptide substrates. Data/citation to support this claim is missing. The figure shows tetrapeptide as a substrate. Please address this.

8) Subsection “Drugs targeting both PBPs and LDTs synergize to kill Mtb”: Clavulanic acid at 5 μg/mL was used for this assay. Therefore, it is important to include this detail here. It is not possible to conclude that amoxicillin and meropenem exhibit synergism as the experiment included another agent (clavulanic acid). Or, the assay needs to be repeated in the absence of clavulanic acid to draw this conclusion. Also, previous work (Kumar P et al., 2017) describes the relationship between various B-lactams and transpeptidases of Mtb. The concluding sentence of this paper is that combinations of two different b-lactams would be most optimal to treat TB. This paper is directly relevant to parts of this manuscript and should be considered.

9) The synergy experiments in *M. tuberculosis* are a (wholly unnecessary) distraction from the importance of the study. It would be far more useful to see the synergy (and or lack of) with the various *M. smegmatis* mutants. None of the other studies were done in Mtb and there is not even an attempt to discuss the relevant enzymes or orthologs in Mtb. Please address this.

10) Currently, the figures are highly complex and the manuscript would benefit from an improved lead in their content. Figure 6 is limited to substrate and enzyme localization. The relation of the central part to panels A and B is unclear. Considering this, Figure 6 is not at the high quality level of the rest of the manuscript. Address this. A related minor point is that Figure 6 is introduced before Figure 3 to 5. As *eLife* caters to a broad readership base, improved flow of your text, clear figures demonstrating key conclusions and a carefully synthesized graphic of the central conclusions would substantively enhance the value of your submission.

[Editors' note: further revisions were requested prior to acceptance, as described below.]

Thank you for resubmitting your work entitled "Maturing *Mycobacterium smegmatis* Peptidoglycan Requires Non-canonical Crosslinks to Maintain Shape" for further consideration at *eLife*. Your revised article has been favorably evaluated by Wendy Garrett (Senior Editor), a Reviewing Editor, and one reviewer.

The manuscript has been improved but there are some remaining issues that need to be addressed before acceptance, as outlined below:

1) With regards to the previous requests to correct the mass spectroscopy data, the ion designation is still incorrect [M+H] 1+ and [M+2H] 2+. Should the m/z ratio in panel C not be 1012.469?

2) The structure of the ions that correspond to the m/z 903.4 and 904.4 (and 974.5 and 975.5) should be provided.

3). Neutral mass = calculated monoisotopic mass. Observed mass [M+H] should be replaced by the observed monoisotopic mass (or observed m/z [M+H] 1+ if preferred). m/z should be italicized. A brief explanation of the techniqueshould be introduced in the legend.

---

## [Author Response]

Essential revisions:1) In Figure 1B, how do the authors determine the new from the old pole? This most likely related to birth size and intensity of staining at the cell pole, but some illustration of this would be useful in the figure and is required to understand the rest of the work.

We thank the reviewers for this suggestion. We performed time-lapse microscopy with both FDAA and Alexa488 NHS ester to determine which pole stains brighter with FDAA and from which pole the gradient originates. These data can now be found in Figure 1—figure supplement 2 and are mentioned in the first paragraph of the subsection “Fluorescent D-amino acids are incorporated asymmetrically by L,D99 transpeptidases”.

2) Data demonstrating how the genotype of the Msm strains lacking all 6 (and 3) LDT-encoding genes was verified is necessary. It could be Southern blotting or sequencing. Sequencing is mentioned in the Materials and methods section, but it is not clear what was done to ensure that there were no unintended cross-over events elsewhere in the chromosome as well.

We performed whole genome sequencing on wild-type mc^2^155 as well as the ΔLDT mutant. Sequencing was done on an Illumina HiSeq 4000 with 150 bp paired-end reads. There was a mean depth of coverage of 148x.

All 6 LDT genes were verified as deleted. Furthermore, there was no evidence of any duplications or cross-over events based on a coverage plot.

As often occurs with strains that have been manipulated extensively, there were unrelated SNPs between the strains (listed below).

–MSMEG_0210:G328V (lprO)

–MSMEG_0946:T35T

–MSMEG_3529:S204S

–MSMEG_3627:M76I (ureA)

–MSMEG_4303:V283D (methyltransferase)

–MSMEG_5447:+c in aa 28/516 (dolichyl-phosphate-mannosyltransferase)

–MSMEG_6454:D190A

The sequencing has been uploaded to NCBI’s SRA (details for sample identifiers are provided below).

STUDY: PRJNA451029 (SRP141343)

ΔLDT SAMPLE: deltaLDTAEBCGF (SRS3442031)

ΔLDT EXPERIMENT: deltaLDTAEBCGF (SRX4275943)

ΔLDT RUN: deltaLDTAEBCGF_R2.fastq (SRR7403831)

WT SAMPLE: Msmeg-KB (SRS3442032)

WT EXPERIMENT: Msmeg-KB (SRX4275944)

WT RUN: Msmeg-KB_R2.fastq (SRR7403830)

These data are mentioned in the subsection “3-3 crosslinks are required for rod shape maintenance at aging cell wall” and methods were added in the subsection “Expression and purification of MSMEG_2433 (DacB2)”).

3) Why just complement with LDTE? Were the other three LDTs enriched/identified in the screen? Was the sextuple mutant necessary to completely eliminate FDAA staining? In this context, for rigour, genetic complementation of the LDT triple mutant is necessary, this seems relatively easy. The same construct used to complement the ΔLDT mutant can be used in this case.

Our screen predicted that the loss of *LDTE* would result in the most substantial loss of FDAA incorporation. For this reason, we chose to complement with *LDTE* alone. As the reviewer suggested, we expressed the same construct we have used to complement the phenotypes of the sextuple ΔLDT mutant, in the triple Δ*LDTAEB* mutant, and found that it partially restores the loss of FDAA incorporation in that genetic background as well. These data are shown in Figure 1—figure supplement 3A and mentioned in the last paragraph of the subsection “Fluorescent D-amino acids are incorporated asymmetrically by L,D-transpeptidases”). These data suggest that additional expression of either *LDTA* or *LDTB* may be required for full complementation, something we have not tested.

Additionally, FDAA staining in the sextuple mutant is roughly three fold lower than in the Δ*LDTAEB* mutant, showing that deletion of additional *LDT* genes leads to additional reduction in FDAA staining.

4) The study unambiguously shows that DacB2 has endopeptidase activity (cleavage of 4-3 crosslinks) and that DacB2 co-localizes with LDTE. The authors conclude that the endopeptidase activity of DacB2 is responsible for the formation of blebs in the absence of LDTs. Direct evidence of the proposed model should be obtained by constructing a dacB2 ΔLDT mutant. As dacB2 was deemed to be nonessential in genome wide inactivation studies in M. tuberculosis and direct gene deletion (Microb Pathog. 2012 52:109-16), this experiment should be feasible and is required to support the conclusions of the study. Genetic complementation of the resulting combinatorial mutant (dacB2 ΔLDT mutant) is also required.

We thank the reviewers for suggesting this experiment. We have attempted to knock-out *dacB2* in the ΔLDT strain, but were unsuccessful. The biological meaning of this is unclear, as the ΔLDT strain already grows quite slowly, and even more so when expressing the RecET recombination machinery. To address this same point, we used a recently-developed mycobacterial CRISPRi system. In the ΔLDT background, we constructed a strain that expresses both a *dacB2*-specific guide RNA and dCas9 upon addition of anhydro-tetracycline (aTc). Addition of aTc led to smaller blebs compared to the blebs formed by the same strain not induced by aTc (Figure 4—figure supplement 3). These data are consistent with our model, in which bleb formation in ΔLDT is due to unchecked endopeptidase activity. However, as knockdown of *dacB2* did not completely eliminate bleb formation in ΔLDT, we hypothesize that there are additional *M. smegmatis* D,D-endopeptidases. The text for this experiment can be found in the second paragraph of the subsection “Peptidoglycan synthesizing enzymes localize to differentially aged cell wall”.

5) The MS analysis of peptidoglycan structure appearing in "Figure 1——figure supplement 3B" raises numerous issues including the absence of any definition of the molecular ions that were detected and the inversion of the donor and acceptor stems between the sets of 4-3 and 3-3 crosslinked dimers. There are inconsistencies between the observed mass and the proposed structures, e.g. peaks 7 and 14 display the same mass but contain 2 and 4 NH2, respectively. Please, be consistent and list the number of NH2 rather than a mixed designation with OH for certain peaks. More importantly, peak 10 disappeared upon deletion of the LDT genes although it is reported to contain a 4-3 crosslink (formed by PPBs). These shortfalls must be addressed.

We have re-analyzed these data and created a new, more informative and more clearly labeled figure (Figure 1—figure supplement 4). We have provided EICs (extracted ion chromatograms) for 3-3 crosslinked species that are readily detectable. We do not detect any significant 3-3 crosslinks in the ΔLDT strain. The disappearance of peak 10 is curious. However, since we have deleted 6 cell wall enzymes in this strain, we expected there to be other changes in peptidoglycan besides disappearance of 3-3 crosslinks. For instance, the LDTs may play important roles as partners in complexes responsible for creating specific 4-3 crosslinks.

6) The experiments showing the cellular localization of the various proteins are not really as clear. Figure 4, subsection “Peptidoglycan synthesizing enzymes localize to differentially aged cell wall”, first paragraph: While it is clear that LDTE-mRFP localized farther from the poles, it appears PonA1-RFP localizes throughout the cell wall as the staining is clearly visible throughout the cell.Subsection “Peptidoglycan synthesizing enzymes localize to differentially aged cell wall”, last paragraph: Figure 4A, bottom panel, shows prominent staining in the poles as well. Therefore, the conclusion that 'DacB2-mRFP localized closer to LDT-mRFP' is not entirely supported by the data.Similarly, from the Discussion (end of third paragraph) to conclude that 3-3 cross-linkages are responsible for localizing the biosynthetic machinery seems inadequate from the location of only one of these proteins. These shortfalls can be addressed by attempting localization of another LDT homologue and providing clearer pictures of how PonA1-RFP localizes. Please ensure that any quantification of localization intensity is done for a sufficient number of cells to ensure robustness.

Upon rereading this section, we agree with the reviewers that it is not written clearly and we have carefully rewritten this section (subsection “Peptidoglycan synthesizing enzymes localize to differentially aged cell wall”). There is a substantial amount of heterogeneity in the localization of all three enzymes. We agree that, for certain cells, PonA1 is localized more uniformly than for other cells. To account for this heterogeneity, we have quantified and averaged the distributions for approximately 25 cells for each strain. This quantification (Figure 4B) clearly shows that PonA1 is most abundant near the poles. While we have not rigorously explored the underlying source of the heterogeneity in the localization of these enzymes, it appears that it could be dynamic in nature. For this reason, we have provided videos of the time-lapse microscopy for these localization studies as well as a frame-by-frame distribution of PonA1-RFP, LDTE-mRFP and DacB2-mRFP in the figure supplement (Figure 4—figure supplement 1). Lastly, we have attempted to localize LDTA-mRFP and LDTB-mRFP using the same cloning scheme as LDTE-mRFP without success.

7) Figure 6A, subsection “Mycobacteria are hypersensitive to PBP inactivation in the absence of LDTs”, first paragraph: It is proposed that D,D-endopeptidase generates tetrapeptide substrates. Data/citation to support this claim is missing. The figure shows tetrapeptide as a substrate. Please address this.

The figure has been modified to clearly show that the 4-3 crosslink generated by the PBP is the substrate for the D,D-endopeptidase reaction. Our in vitro DacB2 experiment (Figure 4C, Figure 4—figure supplement 2) shows that 4-3 crosslinks are cleaved and result in tetrapeptides. As one peptide stem participating in a 4-3 crosslink is a tetrapeptide, when the crosslink is cleaved there must remain a tetrapeptide.

8) Subsection “Drugs targeting both PBPs and LDTs synergize to kill Mtb”: Clavulanic acid at 5 μg/mL was used for this assay. Therefore, it is important to include this detail here. It is not possible to conclude that amoxicillin and meropenem exhibit synergism as the experiment included another agent (clavulanic acid). Or, the assay needs to be repeated in the absence of clavulanic acid to draw this conclusion. Also, previous work (Kumar P et al., 2017) describes the relationship between various B-lactams and transpeptidases of Mtb. The concluding sentence of this paper is that combinations of two different b-lactams would be most optimal to treat TB. This paper is directly relevant to parts of this manuscript and should be considered.

We apologize for missing this critical literature citation and thank the reviewers for this information. We have cited this important work in the Introduction, and in the subsection “Drugs targeting both PBPs and LDTs synergize to kill Mtb Mycobacterium tuberculosis”.

Because clavulanic acid is maintained identically in all conditions, it is controlled for in this experiment and therefore is unlikely to be a confounder for these results.

9) The synergy experiments in M. tuberculosis are a (wholly unnecessary) distraction from the importance of the study. It would be far more useful to see the synergy (and or lack of) with the various M. smegmatis mutants. None of the other studies were done in Mtb and there is not even an attempt to discuss the relevant enzymes or orthologs in Mtb. Please address this.

We thank the reviewers for this important suggestion and agree that, as the manuscript is currently organized, there is a leap to Mtb experiments. We have addressed this by including experiments in Msm (Figure 5A, Figure 5—figure supplement 2) and also by including relevant information regarding LDTs in Mtb (subsection “Drugs targeting both PBPs and LDTs synergize to kill *Mycobacterium tuberculosis”*).

We do feel, however, that direct testing in Mtb is worthwhile as this has consequences for drug therapy. We already know that there are differences in essentiality of LDTs in Msm and Mtb (Kieser et al.,2015); thus, the lack of synergy by MIC in Msm does not necessarily reflect the potential of this treatment for tuberculosis. The kinetic data shown in Figure 5 (and Figure 5—figure supplement 2) illustrates that the combination of these drugs kills Msm more rapidly than either drug alone. Given this, we reasoned that it was necessary to attempt these drug synergy experiments in the pathogen itself.

10) Currently, the figures are highly complex and the manuscript would benefit from an improved lead in their content. Figure 6 is limited to substrate and enzyme localization. The relation of the central part to panels A and B is unclear. Considering this, Figure 6 is not at the high quality level of the rest of the manuscript. Address this. A related minor point is that Figure 6 is introduced before Figure 3 to 5. As eLife caters to a broad readership base, improved flow of your text, clear figures demonstrating key conclusions and a carefully synthesized graphic of the central conclusions would substantively enhance the value of your submission.

The manuscript has been heavily edited to better assist the reader with the figures. We agree that the central panel of Figure 6 (a summary of the left panel) is not necessary, and actually is quite confusing. We have re-made Figure 6 to address these points.